# FRAM: Frobenius-Regularized Assignment Matching with Mixed-Precision Computing

Binrui Shen[1,2], Yuan Liang[3], and Shengxin Zhu[*4,5]

[1]School of Mathematical Sciences, Laboratory of Mathematics and Complex Systems, MOE, Beijing Normal University, Beijing 100875, P.R. China
[2]Faculty of Arts and Sciences, Beijing Normal University, Zhuhai 519087, P.R. China
[3]School of Mathematical Sciences, Beijing Normal University, Beijing 100875, P.R. China
[4]Research Centers for Mathematics, Advanced Institute of Natural Science, Beijing Normal University, Zhuhai 519087, P.R. China
[5]Guangdong Provincial Key Laboratory of Interdisciplinary Research and Application for Data Science, BNU-HKBU United International College, Zhuhai 519087, P.R. China
binrui.shen@bnu.edu.cn, l.y@mail.bnu.edu.cn, Shengxin.Zhu@bnu.edu.cn

## Abstract

Graph matching, usually cast as a discrete Quadratic Assignment Problem (QAP), aims to identify correspondences between nodes in two graphs. Since QAP is NP-hard, many methods its discrete constraints by projecting the discrete feasible set onto its convex hull and solving the resulting continuous problem. However, these relaxations inevitably enlarge the feasible set and introduce two errors: sensitivity to numerical scales and geometric misalignment between the relaxed and original feasible domains. To address these issues, we propose a novel relaxation framework to reformulate the projection step as a Frobenius-Regularized Linear Assignment (FRA) problem. This formulation incorporates a tunable regularization term to curb the inflation of the feasible region and ensure numerical scale invariance. To solve the FRA efficiently, we introduce a scaling algorithm for doubly stochastic normalization. Leveraging its favorable computational properties, we design a theoretically grounded, accelerated mixed-precision algorithm. Building on these components, we propose Frobenius-Regularized Assignment Matching (FRAM), which approximates the QAP solution through a sequence of FRA problems. Extensive CPU experiments show that FRAM consistently outperforms all baselines. On GPUs, with mixed precision, FRAM achieves up to a 370× speedup over its FP64 CPU implementation without sacrificing accuracy.

## 1 Introduction

Graph matching aims to find correspondences between graphs that share potential relationships. It can be used in various fields of intelligent information processing, e.g., image similarity detection [33], graph similarity computation [19, 20], knowledge graph alignment [40], autonomous driving [36], vision-language model alignment [30], point cloud registration [10], deep neural network fusion [25], multi-object tracking [14], and COVID-19 disease mechanism study [13]. However, graph matching is an NP-hard discrete optimization problem [32] and computationally prohibitive for large-scale instances.

To scale up the graph matching problem, many relaxation methods were proposed [2, 5, 12, 21, 27, 34]. These methods relax the discrete problem to a continuous domain and then project the

---

[*]Communication author

39th Conference on Neural Information Processing Systems (NeurIPS 2025).

continuous solution back to the original discrete domain. The doubly stochastic projection is a typical representative and frequently used recently [27, 43]. This projection maps the gradient matrix onto the convex hull of the original domain. However, such relaxations inevitably enlarge the feasible region and result in two sources of errors: (1) geometric misalignment between the relaxed and original domains, which undermines the quality of the recovered integer solution; and (2) the loss of numerical scale invariance due to the projections in the nonconvex setting. Such limitations motivate our current research.

Our contributions include

1. **Theoretical results**: We extend the doubly stochastic projection to the FRA by introducing a tunable regularization parameter that controls relaxation-induced distortion. We provide a thorough analysis of how the regularization parameter affects performance, and address the numerical scale sensitivity through an integrated normalization mechanism.

2. **Algorithm**: we propose FRAM, a graph matching algorithm, that solves the QAP approximately by iteratively solving a sequence of FRA problems. Each FRA problem can be solved efficiently by our proposed Scaling Doubly Stochastic Normalization (SDSN). Empirical evaluations show that FRAM achieves superior performance compared to state-of-the-art baselines.

3. **Computing acceleration**: We develop a theoretically grounded mixed-precision implementation of FRAM. Compared to CPU-based double-precision computation, it achieves up to a 370× speedup on an NVIDIA RTX 4080 SUPER GPU across some benchmark problems, with at most a 0.2% drop in accuracy. To the best of our knowledge, this is the first graph matching algorithm built upon a theoretically grounded mixed-precision design.

## 2 Related Works

**Related graph matching algorithms**. We categorize related graph matching algorithms into three representative classes. (1) Methods based on doubly stochastic optimization: these works employ continuous relaxations. Graduated Assignment (GA) [12] pioneers this approach via a sequence of linear approximations. Adaptive softassign [34] improves GA by automatically tuning an entropic parameter. IPFP [22] projects gradients onto permutations, while DSN [43] finds the nearest doubly stochastic matrix. DSN is adopted by [27] for gradient projection. (2) Spectral-based methods: these approaches leverage spectral properties. Typical methods [21, 35] recover assignments from the leading eigenvector. The method in [5] adds affine constraints to improve accuracy while maintaining speed. Recent research [16] connects eigenvectors to multiscale structural features. (3) Methods based on optimal transport: these techniques formulate graph matching as an optimal transport problem. GWL [39] measures graph distance via Gromov-Wasserstein discrepancy, while S-GWL [38], a scalable variant, applies a divide-and-conquer strategy to enhance efficiency. For a comprehensive survey, we refer readers to [8, 41].

**Mixed-precision computing** is a sophisticated technique for accelerating computationally intensive applications. It has been successfully applied to solving linear systems [4, 15] and large-scale AI models such as DeepSeek-V3 [24]. However, little attention has been paid to its use in the graph matching context, partly due to a lack of theoretical understanding. The main challenges in mixed-precision computing stem from (1) the limited range of lower-precision formats, which increases the risk of overflow or underflow, and (2) rounding errors introduced by lower-precision operations, which may lead to error propagation [28]. See [17] for more details on mixed-precision algorithms. This paper provides theoretical guarantees for a mixed-precision graph matching algorithm, achieving both numerical stability and computational speedups.

## 3 Preliminaries

An *undirected attributed graph* $G = \{V, E, A, F\}$ consists of a finite set of nodes $V = \{1, \ldots, n\}$ and a finite set of edges $E \subset V \times V$. The matrix $A$ is a nonnegative symmetric *edge-attribute matrix* whose element $A_{ij}$ specifies the attribute of the edge between nodes $i$ and $j$. The $i$-th row of the feature matrix $F$ represents the attribute vector of node $i$.

**Matching matrix**. Given two attributed graphs $G = V, E, A, F$ and $\tilde{G} = \tilde{V}, \tilde{E}, \tilde{A}, \tilde{F}$, we first assume that the two graphs have the same number of vertices, i.e., $n = \tilde{n}$, for simplicity. A matching matrix $M \in \mathbb{R}^{n \times n}$ encodes the correspondence between nodes: $M_{i\tilde{i}} = 1$ if node $i$ in $G$ matches node $\tilde{i}$ in $\tilde{G}$, and $M_{i\tilde{i}} = 0$ otherwise. Under the one-to-one constraint, a matching matrix is a permutation matrix. The set of permutation matrices is denoted as $\Pi_{n \times n} = \{M : M\mathbf{1} = \mathbf{1}, M^T\mathbf{1} = \mathbf{1}, M \in \{0, 1\}^{n \times n}\}$, where $\mathbf{1}$ represents vectors with all ones.

**Continuous relaxation**. The graph matching problem is typically formulated as a QAP that is NP-hard [11]. A common strategy to handle such discrete problems is relaxation. It first finds a solution on $\mathcal{D}_{n \times n} := \{N : N\mathbf{1} = \mathbf{1}, N^T\mathbf{1} = \mathbf{1}, N \geq 0\}$ which is the convex hull of the original domain. The relaxed problem is then formulated as

$$N^* = \arg \max_{N \in \mathcal{D}_{n \times n}} \Phi(N), \quad \Phi(N) = \tfrac{1}{2} \underbrace{\operatorname{tr}(N^T A N \widetilde{A})}_{\text{Edges' similarites}} + \lambda \underbrace{\operatorname{tr}(N^T K)}_{\text{Nodes' similarites}}, \tag{1}$$

where $\lambda$ is a parameter, $K = F\tilde{F}^T$, and $\operatorname{tr}(\cdot)$ represents the trace operator. And then $N^*$ is transformed back to the original discrete domain $\Pi_{n \times n}$ by solving a *linear assignment problem*:

$$M = \arg \min_{P \in \Pi_{n \times n}} \|P - N^*\|_F. \tag{2}$$

The matrix $M$ is the final solution. Although relaxation allows the use of continuous optimization, the problem (1) is non-convex and thus remains NP-hard [31]. Hence, existing algorithms typically aim to obtain high-quality approximate solutions within acceptable time. Further discussions on convex relaxations and their constructions can be found in [1, 6, 9].

**The projected fixed-point method**. Many existing methods [5, 12, 22, 27, 34] adopt a similar iterative framework to efficiently approximate the objective in (1):

$$\begin{aligned} N^{(t+1)} &= (1 - \alpha)N^{(t)} + \alpha D^{(t)}, \\ D^{(t)} &= \mathcal{P}(\nabla\Phi(N^{(t)})) = \mathcal{P}(AN^{(t)}\tilde{A} + \lambda K), \end{aligned} \tag{3}$$

where $\alpha$ is a step-size parameter and $\mathcal{P}(\cdot)$ is an operator which maps the gradient matrix onto a certain set. When the solution domain is relaxed to the convex hull of the original domain (i.e., the set of doubly stochastic matrices), a natural choice for $\mathcal{P}(\cdot)$ is the doubly stochastic projection [27, 43]. It finds the closest doubly stochastic matrix to the gradient matrix $\nabla\Phi(N^{(t)})$ in terms of the Frobenius norm:

$$\mathcal{P}_{\mathcal{D}}(X) = \arg \min_{D \in \mathcal{D}_{n \times n}} \|D - X\|_F. \tag{4}$$

The resulting algorithm is the Doubly Stochastic Projected Fixed-Point method (DSPFP) [27].

## 4 Projection to Assignment

We first propose a regularized linear assignment formulation by examining the numerical sensitivity of the doubly stochastic projection, and then analyze how the regularization parameter affects performance.

### 4.1 Doubly stochastic projection to assignment

It can be observed that the solution to the quadratic assignment problem (1) also maximizes $w\Phi(N)$, where $w$ is a positive scaling constant. This reflects the numerical scale-invariant property of the objective. However, the doubly stochastic projection $\mathcal{P}_{\mathcal{D}}(\cdot)$ fails to preserve this property:

$$\mathcal{P}_{\mathcal{D}}(X) \neq \mathcal{P}_{\mathcal{D}}(wX), \qquad \text{for } X \in \mathbb{R}_+^{n \times n}. \tag{5}$$

Since the objective function is non-convex, this sensitivity can cause the projection-based algorithm to converge to different points under different scalings.

To elaborate the sensitivity, consider:

$$\mathcal{P}_{\mathcal{D}}(wX) = \arg \min_{D \in \mathcal{D}_{n \times n}} \|D - wX\|_F = \arg \min_{D \in \mathcal{D}_{n \times n}} \|D - wX\|_F^2. \tag{6}$$

By expanding the quadratic norm, we have
$$\|D - wX\|_F^2 = \langle D, D \rangle - 2\langle D, wX \rangle + \langle wX, wX \rangle, \tag{7}$$
where $\langle \cdot, \cdot \rangle$ represents the Frobenius inner product. Since $\langle wX, wX \rangle$ is independent of $D$, it does not affect the optimization. Thus,
$$\mathcal{P}_{\mathcal{D}}(wX) = \arg \min_{D \in \mathcal{D}_{n \times n}} \langle D, D \rangle - 2w \langle D, X \rangle. \tag{8}$$

As a result, the problem reduces to
$$\mathcal{P}_{\mathcal{D}}(wX) = \arg \max_{D \in \mathcal{D}_{n \times n}} \langle D, X \rangle - \frac{1}{2w} \langle D, D \rangle. \tag{9}$$

$\langle D, X \rangle$ represents an assignment score.

By referring back to the update formula (3), $X$ corresponds to the gradient matrix $\nabla \Phi(N^{(t)})$:
$$\mathcal{P}_{\mathcal{D}}(w \nabla \Phi(N^{(t)})) = \arg \max_{D \in \mathcal{D}_{n \times n}} \langle D, \nabla \Phi(N^{(t)}) \rangle - \frac{1}{2w} \langle D, D \rangle. \tag{10}$$

$$= \arg \max_{D \in \mathcal{D}_{n \times n}} \langle D, AN^{(t)}\tilde{A} + \lambda K \rangle - \frac{1}{2w} \langle D, D \rangle \tag{11}$$

$$= \arg \max_{D \in \mathcal{D}_{n \times n}} \operatorname{tr}(D^T AN^{(t)} \widetilde{A}) + \lambda \operatorname{tr}(D^T K) - \frac{1}{2w} \langle D, D \rangle \tag{12}$$

Intuitively, higher assignment scores in (12) lead to larger objective values (1) during the iterative process. As $w$ increases, the projection process emphasizes optimizing the objective assignment score (1). Conversely, when $w$ decreases, the significance of the objective assignment score diminishes. This finding characterizes how the scaling constant $w$ affects the optimization process.

Motivated by the above observation, we propose converting the uncontrollable scaling constant $w$ of the problem into a controllable modeling parameter $\theta$, as follows. Specifically, we extend the doubly stochastic projection to a Frobenius-regularized linear assignment problem by introducing a modeling parameter $\theta$, leading to the following formulation.

**Theorem 1.** *The solution to the scaled doubly stochastic projection problem*
$$D_X^\theta = \arg \min_{D \in \mathcal{D}_{n \times n}} \|D - \tfrac{\theta}{2} X\|_F^2 \tag{13}$$

*is equivalent to the solution of a Frobenius-regularized linear assignment problem:*
$$D_X^\theta = \arg \max_{D \in \mathcal{D}_{n \times n}} \Gamma^\theta(X), \quad \Gamma^\theta(X) = \langle D, X \rangle - \tfrac{1}{\theta} \langle D, D \rangle. \tag{14}$$

Furthermore, the flexibility of this formulation allows us to normalize the input matrix $X$ by dividing it by $\max(X)$, thereby eliminating the influence of the scaling constant $w$. Consequently, the weighting of the assignment term during optimization is entirely controlled by $\theta$.

## 4.2 Convergence to optimal assignment

To investigate the properties of FRA, we analyze the limiting behavior of $D_X^\theta$ as $\theta$ approaches infinity and 0.

**Theorem 2.** *Let $X \in \mathbb{R}_+^{n \times n}$ and $\mathcal{F}$ be the convex hull of the optimal permutation matrices for the linear assignment problem with $X$. As $\theta \to \infty$, the matrix $D_X^\theta$ converges to a unique matrix $D^* \in \mathcal{F}$, where $D^*$ is the unique solution to $\min_{D \in \mathcal{F}} \frac{1}{\theta} \langle D, D \rangle$.*

This theorem reveals a key advantage of FRA: unlike standard linear assignment solvers [22] that return an optimal permutation and discard others, $D_X^\theta$ approximates a convex combination of all optimal permutations, preserving richer solution information.

**Corollary 1.** *If there is only one optimal permutation, then $D_X^\theta$ converges to the corresponding permutation matrix.*

When $D_X^\theta$ corresponds to the unique permutation matrix, our algorithm becomes equivalent to IPFP [22] and the classical Frank–Wolfe algorithm [37]. In contrast, when multiple optimal permutations exist, our matching algorithm can capture a richer set of high-quality matches than these two methods [22, 37]. However, if $\theta$ is chosen too small, the resulting matrix $D_X^\theta$ may fail to reveal high-quality correspondences, as the following proposition demonstrates.

**Proposition 1.** *As $\theta \to 0$, the matrix $D_X^\theta$ converges to the matrix $\frac{\mathbf{1}\mathbf{1}^T}{n}$.*

### 4.3 Influence of the parameter

To analyze the impact of the parameter, we quantify the solution quality through a distance metric between $D_X^\theta$ and $D_X^\infty$ for a given matrix $X \in \mathbb{R}_+^{n \times n}$. The total assignment score $\langle D^\theta, X \rangle$ scales with the problem size. To enable scale-independent error analysis, we define a *normalized assignment error* to quantify the performance gap:

$$\epsilon_X^\theta = \frac{1}{n} \left( \langle D_X^\infty, X \rangle - \langle D_X^\theta, X \rangle \right) \tag{15}$$

This normalization effectively decouples the approximation error from the problem scale, providing a stable metric to assess the quality of $D_X^\theta$ across different numbers of nodes—like how the *Mean Squared Error* (MSE) serves as a scale-independent alternative to the *Sum of Squared Errors*.

**Proposition 2.** *For a matrix $X \in \mathbb{R}_+^{n \times n}$, the following inequality holds:*

$$\epsilon_X^\theta \leq \tfrac{1}{\theta}. \tag{16}$$

This proposition establishes that the performance gap between $D_X^\theta$ and $D_X^\infty$ is bounded by $1/\theta$. This theoretical bound guarantees the stability and accuracy of our approximation scheme, ensuring that the solution asymptotically approaches optimal performance as $\theta$ becomes sufficiently large.

Figure 1 illustrates how the matrix $D_X^\theta$ evolves as $\theta$ varies. When $\theta$ is small, the matrix entries are nearly uniform. As $\theta$ increases, $D_X^\theta$ progressively approaches a permutation matrix that lies within the original feasible domain of the QAP. This observation demonstrates that a larger $\theta$ effectively suppresses the bias introduced by relaxation. By selecting an appropriate value of $\theta$, the intermediate solution during the matching process remains confined to a relaxed region that stays close to the original feasible domain of graph matching problems.

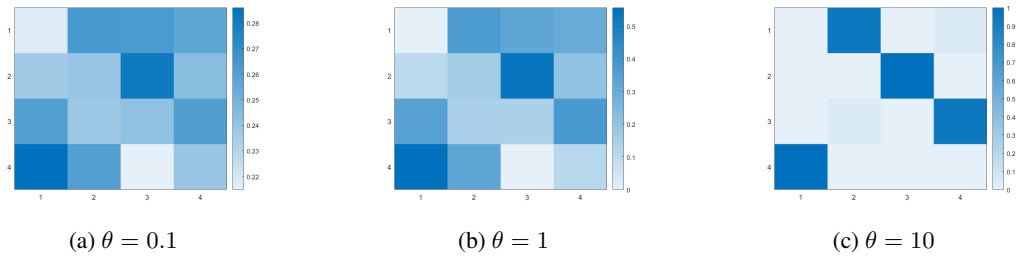

| (a) $\theta = 0.1$ | (b) $\theta = 1$ | (c) $\theta = 10$ |

Figure 1: Visualization of $D_X^\theta$ under different values of $\theta$. The color of each cell represents the matrix entry, with darker shades indicating larger values.

When FRA serves as a component within the graph matching framework (3), increasing $\theta$ generally improves the per-iteration matching score by enforcing sharper correspondences. However, an excessively large $\theta$ may slightly degrade the final matching performance due to premature convergence. This phenomenon can be interpreted from a probabilistic perspective, where each entry in the doubly stochastic matrix $D_X^\theta$ represents the probability of a potential correspondence. A large $\theta$ may lead to overconfident assignments at early stages, thereby hindering the exploration of alternative matching possibilities.

## 5 Scaling Doubly Stochastic Normalization

This section introduces the Scaling Doubly Stochastic Normalization (SDSN) method to efficiently solve the FRA. The method is further adapted to achieve low-precision acceleration, accompanied by theoretical guarantees.

### 5.1 Doubly stochastic normalization

Due to the equivalence between FRA and the scaling doubly stochastic projection, FRA admits a solution via tailored modifications of standard projection algorithms. Zass and Shashua [43] solved

the doubly stochastic projection (4) by alternately solving two subproblems:

$$\mathcal{P}_1(X) = \arg\min_{Y\mathbf{1}=Y^T\mathbf{1}=\mathbf{1}} \|X - Y\|_F, \quad \mathcal{P}_2(X) = \arg\min_{Y\geq 0} \|X - Y\|_F^2 \qquad (17)$$

The von-Neumann successive projection lemma [29] states that $\mathcal{P}_2\mathcal{P}_1\mathcal{P}_2\mathcal{P}_1\ldots\mathcal{P}_2\mathcal{P}_1(X)$ converges to $\mathcal{P}_\mathcal{D}(X)$. The derived doubly stochastic normalization (DSN) [43] for $X \in \mathbb{R}_+^{n\times n}$ works as follows.

$$\tilde{X}^{(k)} = \mathcal{P}_1\left(X^{(k-1)}\right), \;\; X^{(k)} = \mathcal{P}_2\left(\tilde{X}^{(k)}\right), \qquad (18)$$

$$\mathcal{P}_1(X) = X + \left(\frac{I}{n} + \frac{\mathbf{1}^TX\mathbf{1}}{n^2}I - \frac{\mathbf{X}}{n}\right)\mathbf{1}\mathbf{1}^T - \frac{\mathbf{1}\mathbf{1}^T\mathbf{X}}{n}, \qquad \mathcal{P}_2(X) = \frac{X + |X|}{2}, \qquad (19)$$

where $I$ is the $n \times n$ identity matrix. It alternately applies row and column normalization $\mathcal{P}_1$ and non-negativity enforcement $\mathcal{P}_2$ for the doubly stochastic property. Each iteration requires $O(n^2)$ operations. The DSN algorithm converges linearly, meaning that there exists a constant $0 < c < 1$ such that $c = \lim_{k\to\infty} \frac{\|X^{(k+1)}-X^*\|}{\|X^{(k)}-X^*\|}$.

## 5.2 Convergence criterion

An explicit convergence criterion is notably absent in DSN. Zass and Shashua [43] terminated the iterations once the updated matrix became doubly stochastic. However, this approach is computationally expensive and thus inefficient for large-scale tasks. In contrast, Lu et al. [27] fixed the number of iterations to 30 to improve efficiency, but this heuristic did not guarantee that the output matrix remained doubly stochastic.

We propose a criterion to quantify the deviation between the current matrix and the ideal solution. To ensure dimension-invariant analysis and computational efficiency, we define a **dimensional scaled error** as

$$\gamma(X^{(k)}) = \frac{1}{n}\sum_{i,j} X_{ij}^{(k)} - 1 = \frac{\mathbf{1}^TX^{(k)}\mathbf{1}}{n} - 1. \quad (20)$$

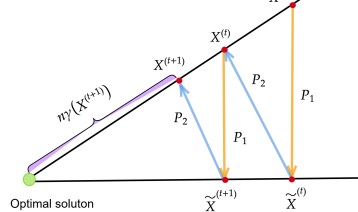

Figure 2: Convergence process of SDSN.

Normalization by $n$ ensures that the metric remains comparable across matrices of different sizes. A detailed derivation of the convergence condition is provided in Appendix B, and the overall process is illustrated in Figure 2.

## 5.3 Number of iterations

The SDSN is summarized in Algorithm 2. We analyze the influence of the parameter $\theta$ on the number of SDSN iterations in the following theorem, which shows that the iteration count grows proportionally with the value of $\theta$.

**Proposition 3.** *For $X \in \mathbb{R}_+^{n\times n}$, the SDSN algorithm requires*

$$\left\lceil \ln\left(\frac{\epsilon}{\theta(\|X\|_F + n)}\right) \frac{1}{\ln(c)} \right\rceil$$

*iterations to produce a solution $X^*$ that satisfies $\|X^* - D_X^\theta\|_F < \epsilon$ where $D_X^\theta$ is the exact solution and $c \in (0, 1)$ is the convergence rate constant of the DSN algorithm.*

## 5.4 Robustness to rounding error

Our method leverages a unique property of SDSN, where the effect of rounding errors diminishes over iterations. This enables theoretically guaranteed low-precision acceleration on modern GPUs. The preservation of accuracy is theoretically guaranteed, ensuring the stability and correctness of the entire process.

**Theorem 3.** *In the SDSN algorithm, the $k$-th iterate $X_k$ can be split as $X_k = \hat{X}_k + \Delta X_k$, where $\Delta X_k$ represents the rounding error. Then, the projection operator satisfies*

$$\mathcal{P}_1(X_k) = \mathcal{P}_1(\hat{X}_k) + \mathcal{P}_3(\Delta X_k),$$

*where $\mathcal{P}_3(\Delta X)$ is the solution to the following minimization problem:*

$$\mathcal{P}_3(\Delta X) = \arg\min_Y \|\Delta X - Y\|_F^2, \quad s.t. \ Y\mathbf{1} = \mathbf{0}, \ Y^\top \mathbf{1} = \mathbf{0}.$$

*The subsequent transformations of the error term $\Delta X_k$ can be expressed as*

$$\mathcal{P}_2\mathcal{P}_3 \cdots \mathcal{P}_2\mathcal{P}_3(\Delta X_k),$$

*which asymptotically converges to the zero matrix $\mathbf{0}_{n\times n}$.*

This property of SDSN prevents the accumulation of rounding errors arising from both the gradient-matrix computation in (3) and the subsequent SDSN operations that correspond to $\Delta X_0$ and $\{\Delta X_k\}_{k\geq 1}$. Therefore, this property enables efficient low-precision computation with negligible loss of accuracy.

## 6 Matching Algorithm

We propose the Frobenius-Regularized Assignment Matching (FRAM) algorithm[2], which approximates the QAP via a sequence of FRA problems. Each FRA is efficiently solved by the scalable SDSN solver. The overall procedure of FRAM is summarized in Algorithm 1.

**Mixed-precision Design on GPU.** The selection of numerical precision for each operation is detailed in inline code annotations. Steps 2-3 perform matrix scaling enabling stable low-precision acceleration in steps 5 and 6. This scaling adjustment is offset by the normalization in SDSN. The subsequent steps are conducted in double precision to compensate for the accuracy degradation. Further implementation details are provided in Appendix D.

**Complexity**. For $n = \tilde{n}$, steps 2–3 require $O(n^2)$ operations. Step 5 requires $O(n^3)$ operations per iteration, regardless of whether fast or sparse matrix computations are used. Step 6 requires $O(ln^2)$ operations where $l$ denotes the number of iterations in SDSN. Step 10 transforms the doubly stochastic matrix $N$ back to a matching matrix $M$ using the Hungarian method [18], which has a worst-case complexity of $O(n^3)$. In practice, the cost is significantly lower because $N$ is sparse in most cases. In short, this algorithm has time complexity $O(n^3 + ln^2)$ per iteration and space complexity $O(n^2)$.

---

**Algorithm 1** Frobenius-Regularized Assignment Matching (FRAM)

**Require:** $A, \tilde{A}, K, \lambda, \alpha, \theta, \delta_{th}$
1: Initial $X^{(0)} = \mathbf{0}_{n\times\tilde{n}}$
2: $c = \max(A, \tilde{A}, K)$ ▷ FP64
3: $A = A/\sqrt{c}, \ \tilde{A} = \tilde{A}/\sqrt{c}, \ K = K/\sqrt{c}$ ▷ FP64
4: **while** $\delta^{(t)} > \delta_{th}$ **do**
5: $\quad X^{(t)} = AN^{(t-1)}\tilde{A} + \lambda K$ ▷ TF32
6: $\quad D^{(t)} = \text{SDSN}(X^{(t)}, \theta)$ ▷ FP32
7: $\quad N^{(t)} = (1-\alpha)N^{(t-1)} + \alpha D^{(t)}$ ▷ FP64
8: $\quad \delta^{(t)} = \|N^{(t)} - N^{(t-1)}\|_F / \|N^{(t)}\|_F$ ▷ FP64
9: **end while**
10: *Discretize $N$ to obtain $M$* ▷ FP64
11: **return** Matching matrix $M$

---

**Algorithm 2** Scaling Doubly Stochastic Normalization (SDSN)

**Require:** Matrix $X, \theta, \gamma_{th}$
1: $X^{(0)} = \frac{\theta}{2}X/\max(X)$
2: **while** $\gamma^{(k)} > \gamma_{th}$ **do**
3: $\quad \bar{X}^{(k)} = \frac{\mathbf{1}^T X \mathbf{1}}{n^2}$
4: $\quad X_1^{(k)} = \left(\frac{\mathbf{I}}{n} + \bar{X}^{(k)}I - \frac{X^{(k)}}{n}\right)\mathbf{1}\mathbf{1}^T$
5: $\quad \tilde{X}^{(k+1)} = X^{(k)} + X_1^{(k)} - \frac{\mathbf{1}\mathbf{1}^T X^{(k)}}{n}$
6: $\quad X^{(k+1)} = (\tilde{X}^{(k+1)} + |\tilde{X}^{(k+1)}|)/2$
7: $\quad \gamma^{(k)} = n\bar{X}^{(k)} - 1$
8: **end while**
9: **Output:** Doubly stochastic matrix $X$

---

[2] https://github.com/BinruiShen/FRAM

# 7 Experiments

We evaluate the proposed algorithm (FRAM) and our additional contributions from the following perspectives:

- **Q1.** What improvements does FRAM deliver over baselines on attributed graph matching tasks?
- **Q2.** How robust is FRAM on attribute-free graph matching tasks?
- **Q3.** How does the mixed-precision design accelerate FRAM?

In addition, we discuss the regularization parameter in Appendix F.

**Setting.** For FRAM, we set $\theta = 2$ for attributed graph matching tasks and $\theta = 10$ for unattributed tasks. Following [27], we set the regularization parameter to $\lambda = 1$ in our experiments, since the results are not sensitive to $\lambda$. We configure $\alpha$ to 0.95 to align with the parameter settings used in DSPFP [27]. All algorithmic comparison experiments are conducted in Python 3 on a workstation equipped with an Intel Core i7 (2.80 GHz) processor. All numerical computations are performed in double precision (FP64) to ensure numerical stability. Typical algorithms such as ASM and GA involve exponential operations and are sensitive to floating-point precision. For evaluating the mixed-precision design, we utilize a hardware platform equipped with an Intel Core i9-14900 (3.20 GHz) CPU and an NVIDIA RTX 4080 SUPER GPU.

**Dataset.** We evaluate our algorithm on three types of graph data: graphs with attributes on both edges and nodes, graphs with attributed edges only, and graphs without attributes. The dataset specifications are summarized in Table 1, and the graph construction procedure from images is described in Appendix E.

| Dataset | $|V|$ | $|E|$ | Attributed nodes | Attributed edges | Ground-truth | Dense graphs |
|---|---|---|---|---|---|---|
| Real-world pictures | (700,1000) | (244 650, 499 500) | ✓ | ✓ | ✗ | ✓ |
| CMU House | (600,800) | (179 700, 319 600) | ✗ | ✓ | ✗ | ✓ |
| Facebook-ego | 4 039 | 88 234 | ✗ | ✗ | ✓ | ✗ |

Table 1: Datasets. $|V|$ is the number of nodes and $|E|$ is the number of edges. ( , ) represents a range.

**Criteria.** For attributed graph matching tasks, we evaluate the matching error of algorithms by

$$\frac{1}{2} \left\| A - M \widetilde{A} M^T \right\|_F^2 + \left\| F - M \tilde{F} \right\|_F^2 . \tag{21}$$

This formulation is mathematically equivalent to the original objective function (1), differing only by an additive constant and a scaling factor. These terms do not affect the optimization process, while the new formulation provides a more intuitive understanding of the problem. For graphs containing only edge attribute matrices, the metric reduces to the first term of (21). For attribute-free graph matching tasks, the metric is defined as $\frac{n_c}{n}$, where $n_c$ denotes the number of correctly matched nodes.

**Baselines** include project fixed-point algorithms such as DSPFP [27] and AIPFP [22, 27]; softassign-based algorithms such as GA [12] (based on (1)) and ASM [34]; optimal transport methods such as GWL [39] and S-GWL [38]; and a spectral-based algorithm, GRASP [16]. Among these, methods based on optimal transport and GRASP are designed for attribute-free graph matching tasks. Most baselines suffer from numerical overflow and accumulation of rounding error under low precision. Therefore, we omit their low-precision comparisons (see Appendix D for details). Many state-of-the-art algorithms, including Path Following [42], FGM [44], RRWM [2], PM [7], BGM [5], and MPM [3], do not scale well to large graphs (e.g., with more than 1000 nodes), and are thus not included in our large-scale graph matching comparisons.

## 7.1 Real-world pictures

In this experiment, the attributed graphs are constructed from a public dataset[3], containing eight sets of pictures. The dataset covers five common transformations: viewpoint change, scale change, image blur, JPEG compression and illumination. The numerical results are presented in Table 2.

---

[3] http://www.robots.ox.ac.uk/~vgg/research/affine/

| Performance | Running Time | | | | | | | |
|---|---|---|---|---|---|---|---|---|
| Image Set | bark | boat | graf | wall | leuv | tree | ubc | bikes |
| DSPFP | 9.1s | 7.3s | 8.8s | 6.1s | 5.8s | 7.6s | 6.1s | 3.3s |
| AIPFP | 44.3s | 44.2s | 84.4s | 44.8s | 26.3s | 40.3s | 34.3s | 16.3s |
| GA | 30.8s | 31.0s | 34.2s | 29.7s | 30.8s | 29.8s | 31.8s | 16.8s |
| ASM | 4.5s | 4.5s | 4.2s | 5s | 5s | 3.8s | 4.5s | 3.2s |
| FRAM | **2.6**s | **2.1**s | **2.4**s | **2.3**s | **2.2**s | **2.5**s | **2.4**s | **1.1**s |

| Performance | Matching Error ($\times 10^4$) | | | | | | | |
|---|---|---|---|---|---|---|---|---|
| Image Set | bark | boat | graf | wall | leuv | tree | ubc | bikes |
| DSPFP | 5.0 | 4.4 | 5.1 | 4.1 | 5.0 | 4.7 | 4.0 | 4.9 |
| AIPFP | 4.6 | 4.5 | 5.3 | 4.4 | 3.9 | 4.7 | 3.6 | 4.3 |
| GA | 4.9 | 5.3 | 6.4 | 6.6 | 4.2 | 6.3 | 3.4 | 4.6 |
| ASM | 4.6 | 4.4 | 4.9 | 4.2 | **3.7** | **3.5** | 3.3 | **3.6** |
| FRAM | **4.2** | **4.0** | **4.6** | **3.6** | 4.9 | 3.8 | **3.1** | 4.4 |

Table 2: Performance comparison in terms of (a) running time and (b) matching error on different image sets. The number of nodes is set to 1000 (*bike* set with 700 nodes). All algorithms are evaluated using double precision (FP64).

As a revolutionary version of DSPFP, FRAM achieves significant acceleration across all image sets. The average runtime of FRAM is 2.3s, compared to 6.5s for DSPFP, yielding an overall speedup of 2.8×. In addition to acceleration, FRAM consistently outperforms DSPFP regarding matching accuracy, demonstrating the effectiveness of the algorithmic design. Overall, FRAM achieves the best matching performance in more than half of the experiments while being nearly twice as fast as the second-fastest method, ASM.

## 7.2 House sequence

CMU house sequence[4] is a classic benchmark dataset. It consists of a sequence of images showing a toy house captured from different viewpoints. Figure 3 demonstrates that FRAM achieves the best performance in both speed and accuracy on the House sequence dataset. It runs 4.1× faster than DSPFP and 3.4× faster than ASM, while attaining the lowest matching error. These results clearly highlight FRAM's efficiency and effectiveness.

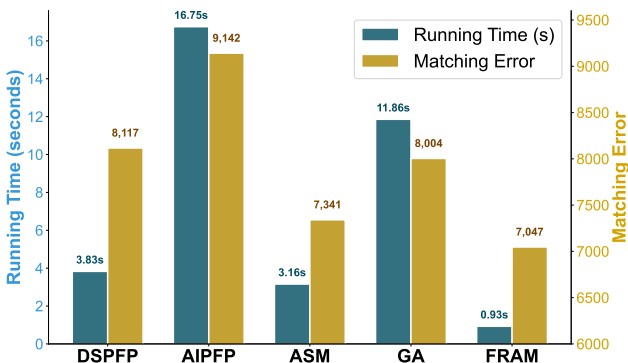

Figure 3: Comparisons between algorithms on graphs from the house sequence. All algorithms are evaluated using double precision (FP64).

## 7.3 Social networks

The social network, comprising 'circles' (or 'friends lists') from Facebook [23], contains 4039 users (nodes) and 88234 relations (edges). We compare different methods in matching networks with noisy versions at 5%, 15% and 25%. Table 3 shows that FRAM achieves the highest node accuracy across all noise levels while maintaining computational efficiency. FRAM achieves 4% higher accuracy than ASM (the second-best method) while running twice as fast. Although FRAM is slightly slower than DSPFP, it offers a substantial 15% improvement in accuracy, demonstrating a favorable trade-off between precision and efficiency.

---

[4]https://www.cs.cmu.edu/afs/cs/project/vision/vasc/idb/images/motion/house/

| Social network | 5% noise | | 15% noise | | 25% noise | |
|---|---|---|---|---|---|---|
| Methods | Acc | Time | Acc | Time | Acc | Time |
| S-GWL | 26.4% | 1204.1s | 18.3% | 1268.2s | 17.9% | 1295.8s |
| GWL | 78.1% | 3721.6s | 68.4% | 4271.3s | 60.8% | 4453.9s |
| DSPFP | 79.7% | 151.3s | 68.3% | 154.2s | 62.2% | 156.9s |
| GA | 35.5% | 793.2s | 21.4% | 761.7s | 16.0% | 832.6s |
| GRASP | 37.9% | **63.6s** | 20.3% | **67.4s** | 15.7% | **71.3s** |
| ASM | 91.1% | 387.2s | 88.4% | 391.7s | 85.7% | 393.1s |
| AIPFP | 68.6% | 2705.5s | 55.1% | 2552.7s | 47.8% | 2513.8s |
| FRAM | **94.7%** | 211.1s | **91.1%** | 221.6s | **89.5%** | 222.9s |

Table 3: Results on Facebook network matching, which are evaluated using double precision (FP64).

## 7.4 Mixed-precision acceleration

This subsection analyzes the acceleration performance of mixed-precision design in FRAM across varying tasks. As demonstrated in Figure 4, the design shows markedly higher acceleration ratios for large-scale problems. Specifically, in the ubc(2000) matching task, mixed-precision design achieves (a) 12.7× speedup over standard GPU-FP64 implementations, and (b) a 371.4× acceleration compared to CPU-FP64 baselines. Conversely, for tasks with up to 1000 nodes, the observed speed-ups are below 4×, likely because the problem scale is insufficient to fully utilize the hardware's computational capacity. These observations are consistent with Amdahl's law: fixed computational overheads dominate runtime at small scales, significantly reducing achievable performance improvements.

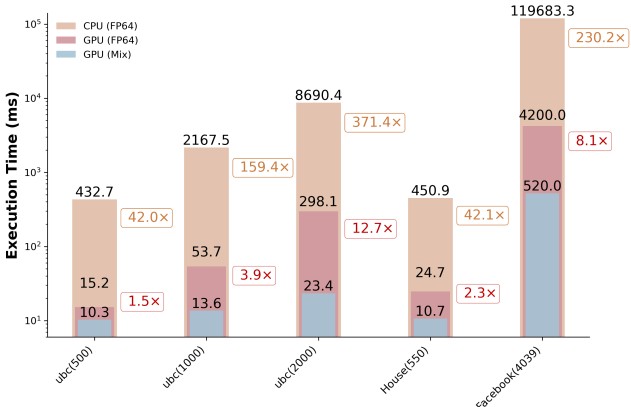

Figure 4: Runtime comparison of FRAM across different precisions and processors. The $y$-axis shows runtime (log scale), and the $x$-axis indicates datasets and their sizes. Red boxed numbers show the speedup of GPU mixed-precision over GPU double precision, and yellow boxed numbers show the speedup over CPU double precision.

## 8 Conclusion

In the context of graph matching, this study investigates the bias introduced by projection-based relaxations. To address this issue, we reformulate the projection step as a regularized linear assignment problem, providing a principled way to control the relaxation error. Building on this formulation, we propose a robust algorithm that achieves competitive accuracy while providing substantial speed advantages over existing baselines, including a significant improvement over the second-best method. On the computational side, we propose a theoretically grounded mixed-precision design. To the best of our knowledge, this is the first theoretically grounded mixed-precision design for graph matching. It achieves significant acceleration while maintaining numerical stability.

A limitation of this study lies in the empirical selection of the parameter $\theta$, so we plan to develop an adaptive parameter selection strategy in future work. While our framework validates the effectiveness of mixed-precision computation, its computational efficiency can be improved. Future work may explore low-level compilation techniques to further optimize the implementation and unlock additional speed gains.

## Acknowledgements

This work was partially supported by the Natural Science Foundation of China (12271047), the National Key Technologies Research and Development Program (2025YFG0202100), and the BNBU research fund (UICR0400008-21; UICR04202405-21). Thanks the support by the Interdisciplinary Intelligence Super Computer Center of Beijing Normal University at Zhuhai and the Guangdong Provincial Key Laboratory of Interdisciplinary Research and Application for Data Science (2022B1212010006).

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

## A  Notations

The common symbols are summarized in Table A.

Table 4: Symbols and Notations.

| Symbol | Definition |
|---|---|
| $G, \tilde{G}$ | matching graphs |
| $A, \tilde{A}$ | edge attribute matrices of $G$ and $\tilde{G}$ |
| $F, \tilde{F}$ | node attribute matrices of $G$ and $\tilde{G}$ |
| $n, \tilde{n}$ | number of nodes of $G$ and $\tilde{G}$ |
| $M$ | matching matrix |
| $\Pi_{n \times n}$ | the set of $n \times n$ permutation matrices |
| $\mathcal{D}_{n \times n}$ | the set of $n \times n$ doubly stochastic matrices |
| $\mathbf{1}, \mathbf{0}$ | a column vector of all 1s,0s |
| $\mathrm{tr}(\cdot)$ | trace |
| $\langle \cdot, \cdot \rangle$ | Frobenius inner product |
| $\| \cdot \|_F$ | Frobenius norm |
| $\theta$ | the parameter in FRA |
| $\alpha$ | the step-size parameter |
| FP8/16/32/64 | 8/16/32/64-bit Floating Point |
| TF32 | TensorFloat 32 |

## B  Convergence Criterions

### B.1  Convergence criterion of SDSN

We recall the SDSN projection steps below:

$$\tilde{X}^{(k)} = \mathcal{P}_1\left(X^{(k-1)}\right), \ \ X^{(k)} = \mathcal{P}_2\left(\tilde{X}^{(k)}\right),$$

$$\mathcal{P}_1(X) = X + \left(\frac{I}{n} + \frac{\mathbf{1}^T X \mathbf{1}}{n^2}I - \frac{\mathbf{X}}{n}\right)\mathbf{1}\mathbf{1}^T - \frac{\mathbf{1}\mathbf{1^T X}}{n}, \qquad \mathcal{P}_2(X) = \frac{X + |X|}{2}.$$

Before the non-negative projection, the updated matrix satisfies

$$\sum_{i,j} \tilde{X}_{ij}^{(k)} = \sum_{\tilde{X}_{ij}^{(k)} > 0} \tilde{X}_{ij}^{(k)} + \sum_{\tilde{X}_{ij}^{(k)} < 0} \tilde{X}_{ij}^{(k)} = n. \tag{22}$$

After applying the non-negative projection (all negative elements are set to 0), we obtain

$$\sum_{i,j} X_{ij}^{(k)} = \sum_{\tilde{X}_{ij}^{(k)} > 0} \tilde{X}_{ij}^{(k)} = n + \sum_{\tilde{X}_{ij}^{(k)} < 0} |\tilde{X}_{ij}^{(k)}|. \tag{23}$$

Since $\sum_{i,j} X_{ij}^{(k)}$ converges to $n$,

$$\sum_{\tilde{X}_{ij}^{(k)} < 0} |\tilde{X}_{ij}^{(k)}| = \sum_{i,j} X_{ij}^{(k)} - n \tag{24}$$

is natural to represent the residual. Moreover, the von Neumann successive projection lemma [29] guarantees that this distance decreases monotonically over successive iterations. Therefore, we define a **dimensional scaled error** as

$$\gamma(X^{(k)}) = \frac{1}{n}\sum_{i,j} X_{ij}^{(k)} - 1 = \frac{\mathbf{1}^T X^{(k)} \mathbf{1}}{n} - 1, \tag{25}$$

where normalization by $n$ ensures applicability to matrices of any size. Furthermore, since $\mathbf{1}^T X^{(k-1)} \mathbf{1}$ is computed in $\mathcal{P}_1(\cdot)$, we can avoid the computation for the error by approximately using $\gamma(X^{(k-1)})$ in $k$-th iteration.

## B.2 Convergence criterion of the matching algorithm

The matching algorithm in DSPFP [27] adopts the following original convergence criterion:

$$\max\left(\left|\frac{N^{(t)}}{\max(N^{(t)})} - \frac{N^{(t-1)}}{\max(N^{(t-1)})}\right|\right), \tag{26}$$

which was also adopted by the ASM [34]. This metric primarily captures local fluctuations between the successive iterations. To better characterize the global structural evolution, we adopt the following **normalized Frobenius criterion**:

$$\delta^{(t)} = \frac{\|N^{(t)} - N^{(t-1)}\|_F}{\|N^{(t)}\|_F}. \tag{27}$$

This criterion demonstrates enhanced suitability for our framework due to two principal considerations: (1) The Frobenius norm inherently aligns with the FRA and the objective formulation, ensuring mathematical consistency throughout the optimization process. (2) The normalization scheme provides a scale-invariant measurement of the variation of the solution trajectory.

## C  Proofs in Section 4

**Theorem 1.** *The solution to the scaled doubly stochastic projection problem*

$$D_X^\theta = \arg\min_{D \in \mathcal{D}_{n \times n}} \|D - \tfrac{\theta}{2}X\|_F^2 \tag{28}$$

*is equivalent to the solution of a Frobenius-regularized linear assignment problem:*

$$D_X^\theta = \arg\max_{D \in \mathcal{D}_{n \times n}} \Gamma^\theta(X), \quad \Gamma^\theta(X) = \langle D, X \rangle - \tfrac{1}{\theta}\langle D, D \rangle. \tag{29}$$

*Proof.* The squared Frobenius norm can be expanded as:

$$\|D - \tfrac{\theta}{2}X\|_F^2 = \langle D, D \rangle - 2\langle D, \tfrac{\theta}{2}X \rangle + \langle \tfrac{\theta}{2}X, \tfrac{\theta}{2}X \rangle. \tag{30}$$

Since the term $\langle \tfrac{\theta}{2}X, \tfrac{\theta}{2}X \rangle$ is a constant with respect to $D$, it does not affect the optimization. Thus, the objective simplifies to:

$$D_X^\theta = \arg\min_{D \in \mathcal{D}_{n \times n}} \langle D, D \rangle - \theta\langle D, X \rangle. \tag{31}$$

As a result, the problem reduces to:

$$D_X^\theta = \arg\max_{D \in \mathcal{D}_{n \times n}} \langle D, X \rangle - \tfrac{1}{\theta}\langle D, D \rangle. \tag{32}$$

$\square$

**Proposition 1.** *For a nonnegative matrix $X \in \mathbb{R}^{n \times n}$, the following inequality holds:*

$$\epsilon_X^\theta \le \frac{1}{\theta}. \tag{33}$$

*Proof.* Since $D_X^\theta$ is the maximizer of $\Gamma^\theta(X)$, we have

$$\Gamma^\theta(D_X^\theta) \ge \Gamma^\theta(D_X^\infty), \tag{34}$$

where $D_X^\infty$ is a maximizer of the unperturbed problem $\max_{D \in \mathcal{D}_{n \times n}} \langle D, X \rangle$.

Expanding the inequality gives:

$$\langle D_X^\theta, X \rangle - \frac{1}{\theta}\langle D_X^\theta, D_X^\theta \rangle \ge \langle D_X^\infty, X \rangle - \frac{1}{\theta}\langle D_X^\infty, D_X^\infty \rangle. \tag{35}$$

Rearranging terms, we have:

$$\langle D_X^\theta, X \rangle - \langle D_X^\infty, X \rangle \ge \frac{1}{\theta}\left(\langle D_X^\theta, D_X^\theta \rangle - \langle D_X^\infty, D_X^\infty \rangle\right). \tag{36}$$

In particular, the maximum of $\langle D_X^\infty, D_X^\infty \rangle$ is $n$ when $D_X^\infty$ is a permutation matrix. Therefore, we have $\langle D_X^\infty, D_X^\infty \rangle \leq n$ for general cases.

$$\langle D_X^\theta, X \rangle - \langle D_X^\infty, X \rangle \geq \frac{1}{\theta}(\langle D_X^\theta, D_X^\theta \rangle - n). \tag{37}$$

Taking the negative of both sides,

$$\langle D_X^\infty, X \rangle - \langle D_X^\theta, X \rangle \leq \frac{1}{\theta}(n - \langle D_X^\theta, D_X^\theta \rangle). \tag{38}$$

Note that $D_X^\theta \in \mathcal{D}_{n \times n}$ is doubly stochastic, so $1 \leq \langle D_X^\theta, D_X^\theta \rangle \leq n$ (The inner product achieves its maximum value when $D_X^\theta$ is a permutation matrix, while attaining its minimum value under the condition that all elements of $D_X^\theta$ are $\frac{1}{n}$). Therefore,

$$\langle D_X^\infty, X \rangle - \langle D_X^\theta, X \rangle \leq \frac{n-1}{\theta}, \tag{39}$$

which completes the proof. $\qquad \square$

**Theorem 2.** *Let $X \in \mathbb{R}_+^{n \times n}$ and $\mathcal{F}$ be the convex hull of the optimal permutation matrices for the linear assignment problem with $X$. As $\theta \to \infty$, the matrix $D_X^\theta$ converges to a unique matrix $D^* \in \mathcal{F}$, where $D^*$ is the unique solution to $\min_{D \in \mathcal{F}} \frac{1}{\theta}\langle D, D \rangle$.*

*Proof.* We prove the theorem in two steps: (1) $D_X^\infty$ lies on the face $\mathcal{F}$, and (2) $D_X^\infty$ must equal $D^*$.

**Step 1.** Since $D_X^\theta$ is defined as the maximizer of

$$\Gamma^\theta(D) = \langle D, X \rangle - \frac{1}{\theta}\langle D, D \rangle, \tag{40}$$

we have

$$\Gamma^\theta(D_X^\theta) \geq \Gamma^\theta(D^*), \tag{41}$$

which implies

$$\langle D_X^\theta, X \rangle - \frac{1}{\theta}\langle D_X^\theta, D_X^\theta \rangle \geq \langle D^*, X \rangle - \frac{1}{\theta}\langle D^*, D^* \rangle. \tag{42}$$

As $\theta \to \infty$, we have

$$\frac{1}{\theta}\langle D_X^\theta, D_X^\theta \rangle \to 0 \quad \text{and} \quad \frac{1}{\theta}\langle D^*, D^* \rangle \to 0. \tag{43}$$

Taking the limit, we get

$$\lim_{\theta \to \infty} \langle D_X^\theta, X \rangle \geq \langle D^*, X \rangle. \tag{44}$$

Since $D^*$ is the optimal solution to the linear assignment problem, for any $D \in \mathcal{D}_{n \times n}$,

$$\langle D, X \rangle \leq \langle D^*, X \rangle. \tag{45}$$

In particular,

$$\langle D_X^\theta, X \rangle \leq \langle D^*, X \rangle. \tag{46}$$

Combining both inequalities, we obtain

$$\lim_{\theta \to \infty} \langle D_X^\theta, X \rangle = \langle D^*, X \rangle, \tag{47}$$

which means $D_X^\infty$ lies in the face $\mathcal{F}$.

**Step 2.** For each fixed $\theta$, the objective $\Gamma^\theta(D)$ is strictly concave in $D$ due to the quadratic term $-\frac{1}{\theta}\langle D, D \rangle$. Strict concavity ensures that for large $\theta$, the maximizer of $\Gamma^\theta(D)$ is unique.

Suppose, for the sake of contradiction, that $D_X^\infty \neq D^*$. Since both are in $\mathcal{F}$, we have

$$\langle D_X^\infty, X \rangle = \langle D^*, X \rangle. \tag{48}$$

If $D_X^\infty$ were distinct from $D^*$, then as $\theta \to \infty$, the perturbed objective $\Gamma^\theta(D)$ would admit at least two different maximizers ($D_X^\infty$ and $D^*$) with the same objective value. This contradicts the uniqueness guaranteed by strict concavity. Therefore, $D_X^\infty$ must coincide with $D^*$.

Since the entire sequence $D_X^\theta$ is bounded and any convergent subsequence converges to $D^*$, it follows that

$$D_X^\theta \to D^* \text{ as } \theta \to \infty. \tag{49}$$

This establishes the desired convergence. $\qquad \square$

**Proposition 2.** *As $\theta \to 0$, the matrix $D_X^\theta$ converges to the matrix $\frac{\mathbf{11}^T}{n}$.*

*Proof.* As $\theta \to 0$, the matrix $D_X^\theta$ exhibits the limiting behavior:

$$\lim_{\theta \to 0} D_X^\theta = \arg \min_{D \in \mathcal{D}_{n \times n}} \langle D, D \rangle = \sum D_{ij}^2. \tag{50}$$

Since $\sum D_{ij} = n$, $\sum D_{ij}^2$ achieve minimum when all entries are equal to $1/n$. Consequently,

$$\lim_{\theta \to 0} D_X^\theta = \frac{\mathbf{11}^T}{n}. \tag{51}$$

$\square$

**Proposition 3.** *For $X \in \mathbb{R}_+^{n \times n}$, the SDSN algorithm requires*

$$\left\lceil \ln \left( \frac{\epsilon}{\theta(\|X\|_F + n)} \right) \frac{1}{\ln(c)} \right\rceil$$

*iterations to produce a solution $X^*$ that satisfies $\|X^* - D_X^\theta\|_F < \epsilon$ where $D_X^\theta$ is the exact solution and $c \in (0, 1)$ is the convergence rate constant of the DSN algorithm.*

*Proof.*

$$X_{k+1} = \mathcal{P}_1 \mathcal{P}_2(X_k) \tag{52}$$

where $\mathcal{P}_1$ and $\mathcal{P}_2$ denote projection operators maintaining the constraint $X_k \mathbf{1} = (\mathbf{1}^T X_k)^T = \mathbf{1}$. This operation can be explicitly expressed as:

$$X_{k+1} = \mathcal{P}_1 \left( X_k + |X_k| \right) \tag{53}$$

with the absolute value operation applied element-wise to matrix entries. Expanding the projection operators yields the detailed formulation:

$$X_{k+1} = \frac{1}{2}(X_k + |X_k|) + \frac{\mathbf{1}^T [\frac{1}{2}(X_k + |X_k|)] \mathbf{1}}{n^2} \mathbf{11}^T - \frac{1}{n} [\frac{1}{2}(X_k + |X_k|)] \mathbf{11}^T - \frac{1}{n} \mathbf{11}^T [\frac{1}{2}(X_k + |X_k|)] + \frac{1}{n} \mathbf{11}^T. \tag{54}$$

Through vectorization and Kronecker product analysis, we transform the matrix equation into its vector form:

$$\text{vec}(X_{k+1}) = A \, \text{vec} \left( |X_k| + X_k \right) + \frac{1}{n} \text{vec} \left( \mathbf{11}^T \right) \tag{55}$$

where $A = \frac{1}{2}(I - \frac{1}{n}\mathbf{11}^T) \otimes (I - \frac{1}{n}\mathbf{11}^T)$ possesses a spectral radius of $\frac{1}{2}$. Defining the error vector $e_k = \text{vec}(X_k) - \text{vec}(X_*)$ for solution matrix $X_*$, we derive the error propagation relationship:

$$e_{k+1} = A \, \text{vec} \left( |X_k| - X_* \right) + A e_k \tag{56}$$

Applying norm inequalities and leveraging the spectral properties of $A$, we obtain:

$$\|e_{k+1}\| \leq \frac{1}{2} \left( c_k \|e_k\| + \|e_k\| \right) = c \|e_k\| \tag{57}$$

where $c = \sup_{1:k} \{ \frac{c_k + 1}{2} \} < 1$ establishes the linear convergence rate.

Considering scaled initial conditions $\theta X$, the first iteration error becomes $\|e_1\| = \|\theta \cdot e_0\|$. For $k$ iterations with scaling factor $\theta$, the error evolution follows:

$$\|e_k\| = \|\theta \cdot c^k \cdot e_0\| \tag{58}$$

The $\epsilon$-accuracy requirement $\|\theta \cdot c^{k'} \cdot e_0\| \leq \epsilon$ leads to the logarithmic relationship:

$$\ln \theta + k' \ln c + \ln \|e_0\| \leq \ln \epsilon \tag{59}$$

Solving for the required iterations yields:

$$k' \geq \frac{\ln(\epsilon/\theta) + \ln(\epsilon/\|e_0\|)}{\ln c} = \frac{\ln\left(\epsilon/(\theta\|e_0\|)\right)}{\ln c} \tag{60}$$

Given the initial error bound $\|e_0\| = \|X\|_F + n \leq \|X\|_F + n$, we finalize the iteration complexity:

$$k' \geq \frac{\ln\left(\frac{\epsilon}{\theta(\|\mathbf{X}\|_F + n)}\right)}{\ln c} \tag{61}$$

This demonstrates the logarithmic relationship between the scaling factor $\theta$ and the required iterations to maintain solution accuracy, completing the proof. $\square$

**Lemma 1.** *The closed-form solution to the optimization problem*

$$\mathcal{P}_3(X) = \arg\min_{Y} \|X - Y\|_F^2, \text{ s.t. } Y\mathbf{1} = Y^T\mathbf{1} = \mathbf{0} \tag{62}$$

*is given by:*

$$\mathcal{P}_3(X) = X + \left(\frac{1}{n^2}\mathbf{1}^T X\mathbf{1} - \frac{1}{n}X\right)\mathbf{1}\mathbf{1}^T - \frac{1}{n}\mathbf{1}\mathbf{1}^T X. \tag{63}$$

*Proof.* The Lagrangian corresponding to the problem takes the form:

$$L(Y, \mu_1, \mu_2) = \text{tr}(YY^T - 2XY) - \mu_1^T Y\mathbf{1} - \mu_2^T Y^T. \tag{64}$$

By differentiating (64), we can obtain the following:

$$\frac{\partial L}{\partial Y} = Y - X - \mu_1\mathbf{1}^T - \mathbf{1}\mu_2^T, \quad \frac{\partial L}{\partial \mu_1} = Y\mathbf{1}, \quad \frac{\partial L}{\partial \mu_2} = Y^T\mathbf{1}. \tag{65}$$

Thus, let $\frac{\partial L}{\partial Y} = 0$, we have:

$$Y = X + \mu_1\mathbf{1}^T + \mathbf{1}\mu_2^T. \tag{66}$$

Applying $\mathbf{1}$ to both sides of the equation above, we get:

$$0 = X\mathbf{1} + n\mu_1 + \mathbf{1}\mu_2^T\mathbf{1}. \tag{67}$$

Multiplying both sides of (67) by $n$, we get:

$$\mathbf{0} = nX\mathbf{1} + n^2\mu_1 + n\mathbf{1}\mu_2^T\mathbf{1}. \tag{68}$$

Multiplying both sides of (68) by $\mathbf{1}\mathbf{1}^T$, we get:

$$\mathbf{0} = \mathbf{1}\mathbf{1}^T X\mathbf{1} + n\mathbf{1}\mathbf{1}^T\mu_1 + n\mathbf{1}\mu_2^T\mathbf{1}. \tag{69}$$

Subtracting (68) and (69), we obtain:

$$(nI - \mathbf{1}\mathbf{1}^T)X\mathbf{1} + n(nI - \mathbf{1}\mathbf{1}^T)\mu_1 = 0. \tag{70}$$

Solving this equation, we find:

$$\mu_1 = \frac{1}{n}X\mathbf{1} - k_1\mathbf{1}, \quad \mu_2 = -\frac{1}{n}X^T\mathbf{1} - k_2\mathbf{1}, \tag{71}$$

where $k_1, k_2 \in \mathbb{R}$.

Substituting the results into (64), we rewrite the Lagrangian as:

$$L(Y, k_1, k_2) = \text{tr}(FF^T - 2XF) + (\frac{1}{n}X\mathbf{1} + k_1\mathbf{1})^T Y\mathbf{1} + (\frac{1}{n}X^T\mathbf{1} + k_2\mathbf{1})^T Y^T. \tag{72}$$

Finally, setting $\frac{\partial L}{\partial Y} = \frac{\partial L}{\partial k_1} = \frac{\partial L}{\partial k_2} = 0$, we solve for $Y$:

$$Y = X + \left(\frac{\mathbf{1}^T X\mathbf{1}}{n^2}I - \frac{1}{n}X\right)\mathbf{1}\mathbf{1}^T - \frac{1}{n}\mathbf{1}\mathbf{1}^T X, \tag{73}$$

which completes the proof. $\square$

**Theorem 3.** *In the SDSN algorithm, the $k$-th iterate $X_k$ can be split as $X_k = \hat{X}_k + \Delta X_k$, where $\Delta X_k$ represents the rounding error. Then, the projection operator satisfies*

$$\mathcal{P}_1(X_k) = \mathcal{P}_1(\hat{X}_k) + \mathcal{P}_3(\Delta X_k),$$

*where $\mathcal{P}_3(\Delta X)$ is the solution to the following minimization problem:*

$$\mathcal{P}_3(\Delta X) = \arg\min_Y \|\Delta X - Y\|_F^2, \quad s.t. \ Y\mathbf{1} = \mathbf{0}, \ Y^\top \mathbf{1} = \mathbf{0}.$$

*The subsequent transformations of the error term $\Delta X_k$ can be expressed as*

$$\mathcal{P}_2\mathcal{P}_3 \cdots \mathcal{P}_2\mathcal{P}_3(\Delta X_k),$$

*which asymptotically converges to the zero matrix $\mathbf{0}_{n \times n}$.*

*Proof.* In SDSN calculations, the nonnegativity-enforcing part can be computed entirely in low precision, so this error analysis focuses on the first component $\mathcal{P}_1$ (19). To examine the behavior of the rounding error $\Delta X_k$ during iterations, we split $X_k$, the variable at the $k$-th iteration, as

$$X_k = \hat{X}_k + \Delta X_k. \tag{74}$$

Then, $\mathcal{P}_1(X_k)$ becomes

$$\left( \frac{I}{n} + \frac{1^T(\hat{X}_k + \Delta X_k)1 I}{n^2} - \frac{(\hat{X}_k + \Delta X_k)1}{n} \right) 11^T + \hat{X}_k + \Delta X_k - \frac{1}{n}11^T(\hat{X}_k + \Delta X_k) \tag{75}$$

$$= \underbrace{\hat{X}_k + \left( \frac{I}{n} + \frac{1^T\hat{X}_k 1 I}{n^2} - \frac{\hat{X}_k}{n} \right)11^T - \frac{11^T\hat{X}_k}{n}}_{\mathcal{P}_1(\hat{X}_k)} + \underbrace{\Delta X_k + \left( \frac{1^T\Delta X_k 1}{n^2}I - \frac{\Delta X_k}{n} \right)11^T - \frac{11^T\Delta X_k}{n}}_{\Delta T_k}.$$

Lemma 1 establishes that $\Delta T_k = \mathcal{P}_3(\Delta X_k)$, so the subsequent transformation of $\Delta X_k$ can be represented as

$$\mathcal{P}_2\mathcal{P}_3 \cdots \mathcal{P}_2\mathcal{P}_3(\Delta X_k).$$

Through this process, the propagation of this rounding error ultimately converges to $\mathbf{0}_{n \times n}$ to satisfy the constraints imposed by both $\mathcal{P}_2(\cdot)$ and $\mathcal{P}_3(\cdot)$.

$\square$

# D   Details of the Mixed-Precision Design

---
**Algorithm 1** Frobenius-Regularized Assignment Matching (FRAM)

---
**Require:** $A, \tilde{A}, K, \lambda, \alpha, \theta, \delta_{th}$
 1: Initial $X^{(0)} = \mathbf{0}_{n \times \tilde{n}}$
 2: $c = \max(A, \tilde{A}, K)$              ▷ FP64
 3: $A = A/\sqrt{c}, \ \tilde{A} = \tilde{A}/\sqrt{c}, \ K = K/\sqrt{c}$       ▷ FP64
 4: **while** $\delta^{(t)} > \delta_{th}$ **do**
 5:    $X^{(t)} = A N^{(t-1)} \tilde{A} + \lambda K$         ▷ TF32
 6:    $D^{(t)} = \text{SDSN}(X^{(t)}, \theta)$         ▷ FP32
 7:    $N^{(t)} = (1 - \alpha)N^{(t-1)} + \alpha D^{(t)}$      ▷ FP64
 8:    $\delta^{(t)} = \|N^{(t)} - N^{(t-1)}\|_F / \|N^{(t)}\|_F$     ▷ FP64
 9: **end while**
10: *Discretize $N$ to obtain $M$*           ▷ FP64
11: **return** Matching matrix $M$

---

**Implementation details of the mixed-precision design.**

Algorithm 1 employs mixed-precision design to improve computational efficiency while maintaining numerical stability, with precision for each operation specified via inline annotations. It starts by

normalizing matrices $A$, $\tilde{A}$, and $K$ in steps 2-3 using FP64 to prevent overflow and ensure input consistency, enabling subsequent low-precision acceleration.

Step 5 employs TF32 (which is computationally similar to FP16) for critical computations. This strategy strikes a balance between performance and accuracy: it utilizes high-throughput GPU fused multiply-add operations while keeping precision loss within acceptable limits, as proven in Theorem 3. In contrast, further reducing the precision risks numerical instability due to inadequate accuracy.

Step 6 then switches to FP32 to maintain consistency, as FP16 would introduce additional truncation error, reducing accuracy by approximately 5% without increasing iterations. In contrast, FP32 maintains nearly the same number of iterations as FP64, balancing efficiency and precision.

Finally, step 7 reverts to FP64 for high-precision iterative updates, compensating for earlier precision trade-offs. Steps 8 and 10 continue in FP64 to ensure final output accuracy. This precision hierarchy accelerates performance in preconditioned computations while maintaining the reliability required for assignment matching tasks.

**Introduction to different data type.**

| Data Type | Bits | Range | Precision | FLOPs (RTX 4080) |
|-----------|------|-------|-----------|------------------|
| FP32 | 32 | $[-10^{38}, 10^{38}]$ | $10^{-6}$ | 52.2 TeraFLOPS |
| TF32 | 19 | $[-10^{38}, 10^{38}]$ | $10^{-3}$ | 209 TeraFLOPS |
| FP64 | 64 | $[-10^{308}, 10^{308}]$ | $10^{-16}$ | 0.82 TeraFLOPS |

Table 5: Characteristics of FP32, TF32, and FP64 floating-point formats. The listed ranges are approximate, based on the IEEE 754 standard. The FLOPs column reports the theoretical peak performance (in TeraFLOPS) achieved by the NVIDIA RTX 4080 GPU for each format. Here, 1 TeraFLOPS equals $10^{12}$ floating-point operations per second, providing a standard measure of compute performance.

**Why were competing algorithms not evaluated in lower precision settings?** Mixed precision requires careful theoretical justification to ensure robustness against truncation errors and to preserve numerical stability. Not all baseline methods are robust in this regard. In particular, GA, S-GWL, GWL, and ASM all involve exponential operators, which are prone to numerical instability. For example, even in double precision, the default parameter settings in S-GWL can already cause numerical overflow, as documented in Appendix F of [34], making single-precision computation even more problematic. Among the remaining baselines, though AIPFP is comparatively more tolerant of reduced precision, it suffers from slow runtime; GRASP is inherently sensitive to noise. Lowering the precision does not alleviate their main limitations.

# E   Graphs from Images

**The construction** consists of three primary steps. (1) Node extraction: SIFT [26] extracts key points as potential nodes and computes the nodes' attributes; (2) Node selection: select the nodes that exhibit a high degree of similarity (i.e., the inner product of feature vectors) to all candidate nodes of the other graph. (3) Edge attribute calculation: nodes form a fully connected graph weighted by inter-node Euclidean distances.

**Real-world images.** The graphs are constructed as described in the previous paragraph. The number of nodes is set to 1000 (For the *bike* set, we only record the results for the first three pictures with 700 nodes, as the other images lack sufficient keypoints.). The running time and matching error are computed by averaging the results over five matching pairs (1 vs. 2, 2 vs. 3, ..., 5 vs. 6) from the same image set.

**House sequence** is a widely used benchmark dataset consisting of 111 grayscale images of a toy house taken from different viewpoints. The graphs are constructed as described in the previous paragraph, but without node attributes, to evaluate the algorithms from a different perspective. Matching pairs consist of the first image and subsequent images with 5 image sequence gaps (such as image 1 vs. image 6 and so on).

# F Discussion of the regularization parameter

We evaluated the impact of the parameter $\theta$ on the FRAM algorithm across different types of tasks, as shown in Table 6. Overall, the algorithm's performance increases as the parameter grows, but the improvements follow a pattern of diminishing marginal effect, while an excessively large $\theta$ may lead to a slight degradation in performance. For tasks with attributed information, relatively small values of $\theta$ are sufficient to achieve good performance. In contrast, for more challenging tasks without attributed information, larger values of $\theta$ are required to achieve the best performance.

| **House** (attributed edges) | | | **Bark** (attributed edges and nodes) | | | **Facebook Network** | | |
|---|---|---|---|---|---|---|---|---|
| $\theta$ | Time (sec) | Error ($\times 10^3$) | $\theta$ | Time (sec) | Error ($\times 10^4$) | $\theta$ | Time (sec) | Node Accuracy |
| 0.1 | 0.4 | 16.4 | 0.1 | 0.5 | 5.4 | 1 | 165 | 69.26% |
| 0.5 | 0.6 | 7.3 | 0.5 | 0.6 | 4.1 | 5 | 192 | 85.92% |
| 1 | 0.7 | 7.2 | 1 | 0.9 | 4.2 | 10 | 222 | 91.69% |
| 2 | 0.9 | 7.0 | 2 | 1.1 | 4.2 | 15 | 246 | 93.48% |
| 4 | 1.2 | 7.1 | 4 | 1.2 | 4.3 | 20 | 263 | 93.98% |

Table 6: Influence of the parameter $\theta$ on algorithmic performance across diverse graph types.

