# OpenReview forum: "FRAM:  Frobenius-Regularized Assignment Matching with Mixed-Precision Computing"
_NeurIPS.cc/2025/Conference — NeurIPS 2025 poster_

### Official Review · Reviewer_jPAJ · 2025-06-27

**Clarity:** 3
**Significance:** 3
**Originality:** 4
**Rating:** 5
**Confidence:** 2

**Summary:**

This paper proposes a novel algorithm that approximately solves the NP-hard quadratic assignment problem, a fundamental task for many AI-field applications. The method introduces a scaling factor, which can control the sharpness of relaxed matrix values and enables gradually reaching a solid permutation matrix.
In addition, it proposes a novel convergence criterion to stop the iteration at the right time. All results are theoretically supported and its superiority is experimentally proven, being approximately three times faster than the SOTA fastest method while maintaining comparable accuracy to the SOTA precise method.

**Questions:**

- Could you clarify the items in Weakness 1?
- An error analysis could enhance the value of this work. Is there any possibility?

**Ethical Concerns:**

["NO or VERY MINOR ethics concerns only"]

**Final Justification:**

Please see [my last comments in the rebuttal discussion](https://openreview.net/forum?id=rqn7XLNUAv&noteId=IyaLCUFb0h).

**Limitations:**

Yes

**Paper Formatting Concerns:**

No concerns

**Quality:**

3

**Strengths And Weaknesses:**

## Strength
1. Graph matching is a fundamental problem for the NeurIPS community.
2. The presentation is kind for many readers.
3. The method includes a theorem and its proof.
4. The experimental results revealed that the method actually works faster, while maintaining SOTA performance.


## Weakness
### 1. Although well written, some sections are unclear.

- The reviewer is unsure if normalizing with $n$ in Eq. (12) is appropriate, given the non-linear complexity increase. Despite this, Eq. (12) claims to provide "a stable metric to assess the quality of …" (L142). A more precise justification is needed.

- “the regularization term” (L150) is not very specific.

- In Fig. 3, the use of stacked bar charts makes it difficult to understand. This reviewer suggests using simple bar charts for clarity.

---

> ### Author Rebuttal · Authors · 2025-07-30
>
> We sincerely thank the reviewer for the positive and encouraging feedback. We are glad that the reviewer found the problem of graph matching relevant to the NeurIPS community, and that the presentation was accessible to a broad audience. We also appreciate your recognition of the theoretical contribution and the empirical results demonstrating both efficiency and strong performance. Your support is greatly appreciated.
>
> **Weakness1. (a)**: We thank the reviewer for the insightful comment. The normalization by $n$ in Equation (12) is not related to algorithmic complexity, but rather serves to define a scale-independent assignment error. Since the assignment score scales linearly with $n$, this normalization decouples the error measure from the problem size, allowing fair comparison across graphs of different scales.
>
> This is similar in spirit to the mean squared error (MSE) metric, where the total squared error is normalized by the number of elements to make the result independent of data size. Likewise, our normalized error provides a stable and interpretable measure of solution quality. We will include this clarification in the revised manuscript.
>
> **Weakness1. (b)**: We thank the reviewer for this helpful suggestion. We will revise the manuscript to clarify the meaning of “the regularization term” on line 150 by explicitly stating it as the term $\frac{1}{\theta} \langle D, D \rangle$.
>
>  **Weakness1. (c)**: We thank the reviewer for the helpful suggestion.  Following the reviewer's advice, we will revise the figure to use simple (unstacked) bar charts to improve readability and make the comparison more straightforward.
>
> **Q1.  Error analysis**
> This is a good question. FRAM searches for a solution to the graph matching problem within a doubly stochastic domain and subsequently discretizes this solution back to a permutation matrix. However, the discretization step renders the perturbation analysis of the algorithm extremely challenging. To indirectly assess the noise robustness of FRAM, we can refer to the perturbation analyses of other algorithms that, while potentially less robust, offer more tractable theoretical guarantees.
>
>  - One class of such algorithms is *matching via degree profiles* [Ding et al.(2021)Ding, Ma, Wu, and Xu], which utilizes node centrality for matching. For Erdős-Rényi graphs $G(n, d/n)$, the algorithm can perfectly recover the true vertex correspondence with high probability, provided that the average degree satisfies $d = \Omega(\log^2 n)$ and the two graphs differ by at most a $\delta = O(\log^{-2}(n))$ fraction of edges. For dense and sparse graphs, these bounds can be further improved to $\delta = O(\log^{-2/3}(n))$ and $\delta = O(\log^{-2}(d))$, respectively.
>  -  Another approach is *spectral-based methods*, including the GRASP \cite{hermanns2023grasp} discussed in our manuscript and LiSA [Shen et al.(2025)Shen, Niu, and Zhu]. These spectral algorithms use leading eigenvectors to match nodes. Consequently, the robustness of these methods is approximately equivalent to the robustness of the principal eigenvector. For instance, one result from LiSA is as follows:
> $$
> \tilde{A} = A + E,
> $$where $E$ represents the perturbations. If
> $$
> \sqrt{2}\|E\|_F \leq \frac{\rho}{2},
> $$
> with $\rho$ denoting the eigengap (i.e., the difference between the two largest eigenvalues of $A$), then the following bound holds:
> $$
>     \|\varphi - \tilde{\varphi}\|_2 \leq 2\sqrt{2}.
> $$
> where $\varphi$ and $\tilde{\varphi}$ are leading eigenvectors of $A$ and $\tilde{A}$.
>
> [Ding et al.(2021)Ding, Ma, Wu, and Xu] J. Ding, Z. Ma, Y. Wu, and J. Xu. Efficient random graph match-
> ing via degree profiles. Probability Theory and Related Fields, 179:29–115, 2021.
>
> [Shen et al.(2025)Shen, Niu, and Zhu] B. Shen, Q. Niu, and S. Zhu. Lightning graph matching. Journal of
> Computational and Applied Mathematics, 454:116189, 2025.

---

> > ### Comment · Reviewer_jPAJ · 2025-08-05
> > **The rebuttal fixes my concerns**
> >
> > Thank you for clarifying my concerns.
> > As 2/3 of other reviewers are learning toward acceptance, this reviewer will keep the first rating.

---

> > > ### Author Response · Authors · 2025-08-06
> > >
> > > Thank you for your thoughtful response. We greatly appreciate your support and consideration.

---

### Official Review · Reviewer_6upw · 2025-06-29

**Clarity:** 2
**Significance:** 2
**Originality:** 2
**Rating:** 2
**Confidence:** 4

**Summary:**

This paper considers the problem of solving quadratic assignment problems via the convex relaxation of replacing the set of permutation matrices by the set of doubly stochastic matrices. It focuses on a specific algorithm for solving the resulting problem by first projecting the gradient of the cost function onto the set of doubly stochastic matrices and subsequently forming a convex combination of the projected gradient with the current iterate. In particular, it investigates scaling the cost function by a constant positive factor (which alters the iterates), and considers conducting different parts of the computation in different precision. Numerical results demonstrate that the resulting scheme is faster and yields lower cost functions than the compared competitors.

**Questions:**

- Please detail the setting of the relaxation, and put the presented work into context of the literature on such relaxations  (see 1) above).
- Please explain the novelty in Algorithm 2, comment on why alternating projections instead of algorithms like Dykstra/ADMM with a common stopping criterion like the primal-dual gap was used (see 2) above).
- Please revisit the presentation of the results incorporating the suggestions in 4) above.

**Ethical Concerns:**

["NO or VERY MINOR ethics concerns only"]

**Final Justification:**

I appreciate the authors' response to my concerns and realize that I am an 'outlier' among the reviewers. Yet, I'd like to keep my rating. The authors' response that "the optimization objective we consider is very classic in the graph matching literature, and its properties are often taken as implicit knowledge in the field. [...] The quadratic cost function used in our formulation is non-convex [41]. " still poses a major problem for me. The problem the authors are trying to solve is the discrete problem of minimizing quadratic costs over the set of permutation matrices. As permutation matrices $P$ have the property that $\|P\|_F^2 = n$ (a constant), the original discrete cost function has the property that it is invariant to adding $\alpha \|P\|_F^2$ for arbitrary $\alpha$. By choosing $\alpha$ to be the largest singular value of the original quadratic costs, the objective becomes convex (but still needs to be minimized over the non-convex, discrete set of permutation matrices). Similarly, choosing $\alpha$ to be minus the largest singular value makes the objective concave. To my mind, there is no "natural" choice of $\alpha$, not even $\alpha=0$ because different ways of modelling the same problem could implicitly cause different $\alpha$. After the convex relaxation, i.e., after relaxing the set of permutation matrices to the set of doubly stochastic matrices, the properties of the quadratic costs matter heavily. Citing [41] to claim the costs are non-convex is, in my opinion, not correct. The prior works I cited have exploited the property of identical formulations over the set of permutation matrices giving rise to different continuous relaxations. As the authors propose a new heuristic to solve graph matching problems via a continuous relaxation, it seems to me that the insight about the properties of the cost function needs to be discussed and possibly compared against (including possible follow-up works that I am not familiar with). I think this is particularly important because the term $\alpha \|P\|_F^2$ I discussed above is identical to the second term in (9) for $\alpha = -2/w$ (although I do not see the connection immediately).

Furthermore, not all answers by the authors seem mathematically correct. The authors state that "Indeed, equation (18) differs from the original objective by a constant factor." I disagree. I agree that - up to a constant - (18) is the same as (1) if both terms in (18) had a square. This is what I remarked in my original review. But in general, $\|X\|_F + \|Y\|_F$ is not a strictly monotone rescaling of $\|X\|_F^2 + \|Y\|_F^2$, which would be needed in order to compare algorithms that minimize the objective with squares via a formulation that does not contain the squares.

In summary, although the focus of the paper might be more on mixed-precision computing than on the above discussion, I do not feel comfortable recommending the acceptance of this paper in its current form.

**Limitations:**

yes

**Paper Formatting Concerns:**

no concerns

**Quality:**

2

**Strengths And Weaknesses:**

While the overall problem is interesting and investigations on different ways of solving the relaxation are interesting, the presented work has several structural weaknesses.
1) It does not specify any properties of the cost function. When optimizing quadratic costs over the set of permutation matrices, $||P||_F^2 = n$ is a constant, such that it can be added or subtracted without changing the minimizer. Thus, for the optimization problem over permutation matrices, it does not matter if the costs are convex, concave or neither of the two. Yet, this property heavily affects the type of problem after relaxing the permutation matrices to doubly stochastic matrices, see
- F. Fogel, R. Jenatton, F. Bach, and A. d’Aspremont. Convex Relaxations for Permutation Problems, NeurIPS 2013.
- N. Dym, H. Maron, and Y. Lipman. DS++ - A flexible, scalable and provably tight relaxation for matching problems, ACM Transactions on Graphics (TOG), 2017.
- F. Bernard, C. Theobalt and M. Moeller, DS*: Tighter Lifting-Free Convex Relaxations for Quadratic Matching Problems, CVPR 2018.

 for exemplary works that discuss and exploit this aspect. In case the quadratic costs are (strongly) convex, the particular choice of optimization algorithm should not matter in terms of the solution quality anymore – the minimizer of the relaxed problem is unique and can be computed by a wide range of different convex optimization algorithms. For a sufficiently concave objective, the minimizer over the doubly stochastic matrices will coincide with the minimizer over the permutations, thus making the relaxation exact, but also NP-hard to minimize. These aspects are neglected in the presented paper. It remains unclear if the claim is to solve a convex problem faster, or a non-convex problem heuristically better for common problem instances.

2) Independent of the properties of the cost function, the subproblem of projecting the (scaled) gradient onto the set of doubly-stochastic matrices is a convex problem. This is well-known, yet the alternating projection algorithm (Alg. 2) takes a lot of room in the paper. Results like the projection of a variable onto a 2-dimensional linear subspace (row- and column-sum equal to one) and the projection onto the non-negative orthant are presented as a theorem despite being standard results from convex optimization. Accelerations of the alternating projection method (such as Dykstra’s algorithm) or ADMM/Douglas-Rachford splitting algorithms are not discussed. In this respect, the paper significantly lacks context in the fundamentals of convex optimization.

3) The aforementioned lack of context is also reflected in phrasing several very simple results from convex analysis (such as theorems 1, 2 and 4) as theorems, partially without a proper contextualization such as the considered algorithm for $\theta=\infty$ becoming the Frank-Wolfe Algorithm.

4) In my opinion, the paper is not well-written and confusing in several aspects. For example,
- “the numerical scale invariance of the quadratic objective, which is destroyed under the projections”  is irritating. I do not know what is meant by “numerical scale invariance of the quadratic objective”. The minimizer of any cost function remains unchanged when multiplying the costs by a positive number. If the costs are convex, then the projection does not “destroy” this property. The iterates might change, but still converge to the same point (under suitable conditions/algorithm parameters).
- Writing that DSN finds "the nearest doubly stochastic matrix", and Lu et al. “adapt the DSN so that the projected gradient matrix is doubly stochastic” is irritating. What is the difference? It would be better to formalize this.
- As discussed above, I do not think that inequality (5) is a good argument to say that some scaling invariance is destroyed.
- $D^*$ in Theorem 3 should be defined mathematically. The current description remains a little vague.
- Alg. (2) uses the max(X,0) for the projection onto the non-negative orthant. I recommend using the same in eq. (16). They are of course identical, but having consistency would be clearer.
- In Theorem 6 the term “convergence rate constant” crucially needs to be defined.
- The phrase that  the “truncation residual vanishes over iterations in SDSN” is unclear to me and should be rephrased/detailed. Similarly $\Delta X_k$ should be related to quantities defined in the algorithm. The current version is difficult to follow.
- The costs in eq. (18) are not identical (up to a constant) to the original costs that are minimized. Squares are missing at the Frobenius norms. Have these been included in the numerical experiments?

5) Considering that one of the main technical contributions is rescaling the cost function (so far in a heuristic way/via a hyperparameter as admitted in the conclusions), I consider the contribution to be somewhat limited. The mixed precision computation is interesting, but needs to be detailed. Running competing algorithms at lower precision is – to my mind – a crucial baseline.

---

> ### Author Rebuttal · Authors · 2025-07-30
>
> We sincerely thank the reviewer for the careful reading and many valuable comments. We find there may be some misunderstandings regarding the core idea of our work. To address this, we would first like to clarify the possible confusion. Below, we highlight the reviewer’s comments, with potentially misinterpreted phrases emphasized:
>
> - “It focuses on a specific algorithm for solving the resulting problem by **first projecting the gradient of the cost function onto the set of doubly stochastic matrices and subsequently forming a convex combination of the projected gradient with the current iterate.**”
>
> - “In particular, it investigates **scaling the cost function by a constant positive factor.**”
>
> The algorithm described above corresponds to the DSPFP method instead of our FRAM. A key observation that motivated our work is that the doubly stochastic projection (DSP) step in DSPFP does not preserve the scaling-invariance property of the cost function, which we address in our formulation. By analyzing this mismatch, we show that the projection step in DSP is a special case of a more general FRA ( Frobenius-regularized Linear Assignment) problem. Under this interpretation, scaling the objective function implicitly changes the regularization parameter in the FRA, thereby leading to different solutions (the objective is non-convex).
>
> This insight naturally motivates our main idea: using FRA to systematically approximate solutions to the QAP. Compared to the projection, this approach offers greater flexibility and interpretability. As noted in lines 163–164, increasing $\theta$ typically improves the step-wise objective score, offering a more principled mechanism for controlling optimization behavior. Empirically, our proposed algorithm demonstrates strong performance across multiple benchmarks, validating our design.
>
> Importantly, we emphasize that $\theta$ is a parameter of the FRA formulation, not a simple scaling factor—in fact, we deliberately use a different symbol $w$ in the paper when referring to scaling. Although FRA can be solved as a scaled DSN, the roles they play in the algorithm are fundamentally different.
>
> We hope this clarification helps the reviewer appreciate the novelty and conceptual depth of our work—beyond simply tuning a boring scaling factor.
>
> **W1**:
> We thank the reviewer for the thoughtful comment. The paper currently does not explicitly discuss the properties of the cost function. This is primarily because the optimization objective we consider is very classic in the graph matching literature, and its properties are often taken as implicit knowledge in the field. The baselines ([9,19,24,30])  do not explicitly analyze the properties of the objective. Nonetheless, we agree with the reviewer that incorporating a discussion (including the references recommended by the reviewer) of the cost function’s properties would enhance the theoretical understanding, especially for non-expert readers. We will clarify this aspect in the revision as follows:
> - The set of doubly stochastic matrices forms a convex set. The quadratic cost function used in our formulation is non-convex [41]. Consequently, the overall optimization problem is non-convex.
>
> - Our algorithm heuristically addresses the non-convex problem and performs well on typical instances.
>
> [41] Aflalo, Y., Bronstein, A., & Kimmel, R. (2015). On convex relaxation of graph isomorphism. _Proceedings of the National Academy of Sciences_, _112_(10), 2942-2947.
>
> **W2**: Regarding Theorem 5, we appreciate the reviewer’s concern. Our intent in including it as a formal theorem was to close a research gap that existed in earlier work, where this step was only implicitly assumed but not explicitly derived. In response to the reviewer’s comment, we will revise the manuscript to downgrade Theorem 5 to a Proposition, to better reflect its foundational but non-novel status.
>
> We also appreciate the reviewer’s suggestion to consider acceleration methods. However, at present, it remains unclear whether these methods (e.g., Dykstra’s algorithm, ADMM, or Douglas-Rachford splitting) exhibit robustness to truncation residual (like Theorem 7). Without such robustness, their potential acceleration effect is likely to be inferior to the overall speedup achieved by Alg.2 under a mixed-precision implementation. **This is because the mixed-precision computing accelerates not only the Alg.2, but also the matrix gradient computation ( $O(n^3)$ complexity), potentially achieving more than a 50X speedup (GPU).** A detailed numerical analysis of these acceleration techniques under reduced precision could be a valuable direction, but it may beyond the scope of the paper.
>
> **W3**: Thanks for the comment. In light of the reviewer’s suggestion, we have downgraded Theorems 2 and 4 to propositions. However, we retain Theorem 1 as a central result of the paper, as it provides the theoretical basis for extending the projected fixed-point framework to a Frobenius-regularized Linear Assignment approximation to the Quadratic Assignment Problem (QAP).
>
> We have also incorporated the connection between the case of $\theta =\infty$ and the classical Frank-Wolfe algorithm, as pointed out by the reviewer. That said, we note that in practice, $\theta$ is not set to infinity due to both computational considerations and potential degradation in performance when $\theta$ becomes too large (see lines 164–168 for details).
>
> **W4.1**：The reviewer is correct that “numerical scale invariance of the quadratic objective” refers to the fact that the minimizer of a cost function remains unchanged under positive scaling of the cost values. However, in our setting, the cost function is non-convex, and we empirically observe that scaling the objective may lead the projection-based algorithm to converge to different points. To avoid confusion, we will revise the original phrasing:
>
> 	“the numerical scale invariance of the quadratic objective, which is destroyed under the projections”
> will be replaced with
>
> 	“the numerical scale invariance of the quadratic objective is not preserved by the projections in the non-convex setting”.
>
> **W4.2:** We appreciate the comment. In fact, there is no significant difference between the two statements. Both refer to the use of DSN to project the gradient matrix onto the set of doubly stochastic matrices during the optimization process. We will revise the manuscript to make this clearer.
>
> **W4.3:** We agree that the objective function itself remains invariant under positive scaling. What we intended to convey is that although the minimizer of the objective is unchanged in theory, the intermediate projection steps are sensitive to such scaling. Since the problem is non-convex, this sensitivity can cause the projection-based algorithm to converge to different points under different scalings. We will revise the manuscript to make this clearer.
>
> **W4.4:** Thanks for the suggestion, we will revise the manuscript.
>
> **W4.5:** Thanks for the comment. Equation (16) follows the original DSPFP formulation. Our implementation, as shown in Algorithm (2), includes a minor modification that yields a slight speed improvement. We will emphasize this distinction more clearly.
>
> **W4.6:** We thank the reviewer for this valuable comment. Indeed, the term “convergence rate constant” is already defined in the proof of Theorem 6. To enhance clarity and readability, we will provide a formal definition of this term in the main text.
>
> **W4.7:** Truncation residual refers to the error introduced in numerical calculations when certain parts of a value are truncated (cut off) or approximated due to the limitation in precision. This typically occurs when a high precision value is converted into a lower precision value, resulting in the loss of some information.
>
>  Example: let's say we have a floating point number x = 3.14159265358979, and we need to approximate it by converting it to low precision (for example, 4 decimal places): $x_{\text{low}} = 3.1416$. The truncation residual is the difference between the original value and the truncated value:
> $\text{Truncation Residual} = x - x_{\text{low}} = 3.14159265358979 - 3.1416 = -0.00000734641021$.
>
> **W4.8:** We fully agree with the reviewer’s observation. Indeed, equation (18) differs from the original objective by a constant factor. Since our focus is on the matching matrix rather than the objective value, we chose to follow previous work and used the more interpretable form of equation (18) as the experimental metric. To ensure greater rigor in our manuscript, we will include a clarification of this point in the revised version.
>
> **W5:** We sincerely thank the reviewer for the positive assessment of our use of mixed precision computation. As noted at the beginning of our rebuttal, our approach does not involve “rescaling the cost function,” so we will not repeat that clarification here.
>
> We agree that, in principle, evaluating competing algorithms under lower precision could serve as a valuable baseline. However, mixed precision requires careful theoretical justification to ensure robustness against truncation residual and to preserve numerical stability. Not all baseline methods are robust in this regard. In particular, GA, S-GWL, GWL, and ASM all involve exponential operators, which are prone to numerical instability. For example, even in double precision, the default parameter settings in S-GWL can already cause numerical overflow, as documented in the appendix (Section F) of the ASM [29]. Among the remaining baselines, though  AIPFP is comparatively more tolerant of reduced precision, it suffers from slow runtime; GRASP is inherently sensitive to noise. Lowering the precision does not alleviate their main limitations.
>
> To provide a more complete perspective, we will include in the appendix E (Implementation Details of the Mixed-Precision Architecture) a clarification discussing these precision-related considerations about the baseline methods.

---

### Official Review · Reviewer_tDfZ · 2025-07-01

**Clarity:** 3
**Significance:** 3
**Originality:** 3
**Rating:** 5
**Confidence:** 4

**Summary:**

This work presents a method for graph matching by reformulating projection-based regularizations as linear assignment problems. Doing so, the authors devise an effective and elegant algorithm to control the relaxation. The method is fast without compensating for substantial accuracy. Moreover, the paper introduces a mixed-precision architecture that seems novel and is validated to contribute to the stability of the approach. Overall, this work is a well-written contribution that, although niche in its application, might be interesting to the broader field and could spur further applications beyond graph matching.

**Questions:**

- What are the limits of the precision? At what point does the accuracy break down dramatically?

**Ethical Concerns:**

["NO or VERY MINOR ethics concerns only"]

**Final Justification:**

As discussed post-rebuttal, I did not have many concerns and still support acceptance.

**Quality:**

3

**Strengths And Weaknesses:**

Strength:
- The paper makes a nice theoretical contribution with the formulation of a linear assignment in the projection step for graph matching. The approach is core to a graph matching algorithm by iteratively solving a regularized linear assignment. This is shown to be a fast and effective method.
- The mixed-precision scheme proposed in the paper is a further well-crafted implementation detail that is shown to be effective and well-grounded.
- The manuscript is well written and covers the body of existing work adequately, which allows the authors to situate their work in the landscape of prior work.
- The convergence proofs, although not extensive, are convincing.
- The evaluation leaves no gaps and the experiments are confirming the effectiveness of the method.

Weaknesses:
- While the method may potentially be broadly applicable, the application is limited to graph matching in this manuscript as a rather niche paper.
- Although this is adequately evaluated, there is cost in accuracy.

---

> ### Author Rebuttal · Authors · 2025-07-30
>
> We sincerely appreciate the reviewer’s thoughtful and positive feedback. We are pleased that you found the theoretical contribution—particularly the formulation of the linear assignment in the projection step and the iterative regularized approach—both core and effective. We’re also grateful for your recognition of the mixed-precision scheme as a meaningful implementation detail, as well as your comments on the clarity of the manuscript and its positioning within the existing literature. Your affirmation of the convergence analysis and the thoroughness of our evaluation is especially encouraging. Thank you again for your valuable and encouraging review.
>
> **Weakness1. Broadly applicable:** We are grateful to the reviewer for the high appreciation of our work. While this manuscript focuses on graph matching as a representative application, we plan to explore and extend its applicability to other domains in future work.
>
> **Weakness2. Cost in accuracy:** We thank the reviewer for the careful reading and thoughtful comment. In practice, the mixed-precision version exhibits only a negligible drop in accuracy—at worst, 0.2% lower than the full double-precision variant. Interestingly, due to the final discretization step from the continuous solution, the mixed-precision implementation even slightly outperforms the double-precision version in some cases. We will incorporate these observations into the main text to provide a more complete and balanced evaluation.
>
> **Q1. Precision:**
> We thank the reviewer for raising this important question. Mixed-precision computation indeed introduces two key considerations: numerical stability and error propagation. In our method, low-precision computation is used in two core modules: gradient matrix calculation and SDSN. We address the reviewer’s question from the perspective of precision design in these two components:
>
> - Gradient Matrix Computation: We use TensorFloat-32 (TF32) for this step, which shares a similar precision with FP16. Further reducing the precision beyond TF32 risks numerical instability, especially for large-scale problems, due to insufficient representation accuracy.
>
> - SDSN Module: This component is executed in FP32. While it is technically possible to use FP16, doing so introduces additional truncation residual. Without increasing the number of iterations accordingly, the accuracy drops by approximately 5%. On the other hand, the number of iterations under FP32 is almost identical to that of FP64, offering a good balance between efficiency and precision. Thus, we adopt FP32 as the most reliable choice in this context.
>
> We appreciate the reviewer’s insightful question and will incorporate a detailed discussion of these design choices and precision limitations into the revised manuscript to enhance completeness.

---

> > ### Comment · Reviewer_tDfZ · 2025-08-05
> > **Rebuttal**
> >
> > Thank you for the rebuttal. I did not have many concerns and still would support acceptance.

---

> > > ### Author Response · Authors · 2025-08-06
> > >
> > > Thank you for your feedback and support. I appreciate your positive assessment and am glad to hear there were no major concerns.

---

### Official Review · Reviewer_2pmt · 2025-07-02

**Clarity:** 3
**Significance:** 3
**Originality:** 3
**Rating:** 5
**Confidence:** 3

**Summary:**

This paper presents a novel algorithm for graph matching, reformulating the projection step as a Frobenius-regularized linear assignment (FRA) problem to address the NP-hardness of the Quadratic Assignment Problem (QAP), mitigating errors from feasible region inflation and numerical scale sensitivity via a tunable regularization term. It introduces the Scaling Doubly Stochastic Normalization (SDSN) algorithm to solve FRA efficiently. Furthermore, a theoretically grounded mixed-precision architecture that achieves acceleration on GPU with negligible accuracy loss compared to CPU. The approach outperforms baselines in accuracy and efficiency across various datasets, and provides theoretical analyses of regularization effects, convergence, and error bounds, marking the first graph matching algorithm with a theoretically sound mixed-precision design.

**Questions:**

Q1. It seems that each iteration in Algorithm~2 takes $O(n^2)$ time complexity, and the number of iterations is related to more parameters rather than $n$. So the complexity of Step~6 in Algorithm~1 seems not accurate, which also influences the total time complexity of Algorithm~1.

Q2. What does the matching error refer to? I guess it should be the accuracy defined in Equation~(18), so please use a consistent terminology in the paper.

Q3. What does $K$ refer to in Algorithm~1? I guess it should be the $F\tilde{F}^T$. Please introduce it in Algorithm~1.

**Ethical Concerns:**

["NO or VERY MINOR ethics concerns only"]

**Final Justification:**

Most of my concerns are addressed. Thus, I will increase my rating.

**Limitations:**

yes

**Quality:**

3

**Strengths And Weaknesses:**

Strength:

S1. This paper improves the popular DSPFP methods by reformulating the projection step as an FRA problem, extending the selection of projections such that improve the solution. Corresponding solution, SDNS algorithm, is also proposed.

S2. The paper provides a solid theoretical analysis of the proposed FRA problem and SDSN solution, including the impact of the regularization parameter, convergence properties, and error bounds.

S3. Extensive experiments on real-world datasets demonstrate that FRAM outperforms baselines in both accuracy and speed.

Weakness:

W1. In my comprehension, the proposed FRA problem introduces a family of projection problems, which should lead to equivalent graph matching, but actually results differently due to the projection. In other words, the FRA introduces a family of alternatives in the projection step. Hence, the issue of my most concern is how to select the proper FRA problem, in other words, the parameter $\theta$. However, such strategies are not discussed in the paper.

W2. The paper only conducts the experiments with $\lambda = 1$. It remains unclear whether the value of $\lambda$ will influence the performance of graph matching algorithms.

W3. The $O(n^2)$ space complexity leads to the worry about the scalability of the proposed FRAM method.

---

> ### Author Rebuttal · Authors · 2025-07-30
>
> We sincerely thank the reviewer for the positive and encouraging comments on our work. We are pleased that you recognized our improvements to the DSPFP methods (S1), the solid theoretical analysis provided (S2), and the extensive experiments demonstrating the advantages of our approach on real-world datasets (S3). Your affirmation is truly motivating and strengthens our confidence to continue pursuing research in this direction. Thank you again for your thoughtful review and kind recognition.
>
> **W1:** We thank the reviewer for this insightful comment. The parameter $\theta$ indeed plays a pivotal role: it governs the error bound between the projected matrix $D_X^\theta$ and the optimal linear assignment solution $D_X^\infty$ (Theorem 2), and it also correlates with the number of iterations required by SDSN (Theorem 6). Accordingly, our selection strategy is to choose the smallest feasible value of $\theta$ that preserves matching performance, thereby maximizing computational efficiency. Empirically, when graphs are plain (e.g., unattributed nodes and edges), a larger $\theta$ is necessary; conversely, when attribute information is richer (e.g., edges carry meaningful attributes), a smaller $\theta$ suffices.
>
> We will supplement the manuscript with a dedicated discussion of parameter‐selection strategies for $\theta$, along with comprehensive experimental results. Due to space constraints in this rebuttal, we present below a concise summary of how varying $\theta$ impacts both runtime and matching accuracy.
>
>
> **Plain Graph: Facebook**
> | $\theta$ | Time (sec) | Mean Node Acc |
> | :------: | :--------: | :-----------: |
> |    1     |    165    |    69.26%     |
> |    5     |    192    |    85.92%     |
> |    10    |    222    |    91.69%     |
> |    15    |    246    |    93.48%     |
> |    20    |    263    |    93.98%     |
>
> **Attributed Graph: House**
> |  θ  | Time (sec) | Matching Error($\times 10^3$) |
> | :-: | :--------: | :---------------------------: |
> | 0.1 |    0.4     |             16.4              |
> | 0.5 |    0.6     |              7.3              |
> |  1  |    0.7     |              7.2              |
> |  2  |    0.9     |              7.0              |
> |  4  |    1.2     |              7.1              |
>
> **Attributed Graph: Bark**
> |  θ  | Time (sec) | Matching Error($\times 10^4$) |
> | :-: | :--------: | :---------------------------: |
> | 0.1 |    0.5     |              5.4              |
> | 0.5 |    0.6     |              4.1              |
> |  1  |    0.9     |              4.2              |
> |  2  |    1.1     |              4.2              |
> |  4  |    1.2     |              4.3              |
>
> **W2:** We sincerely thank the reviewer for the careful reading and insightful comment. The parameter $\lambda$ serves to balance the contributions of edge and node information in the graph matching process. As discussed in detail in DSPFP [23] (see lines 244–245), this parameter has already been extensively studied. To maintain focus and avoid redundancy, we did not re-evaluate this parameter.
>
> In line with DSPFP [23], we set $\lambda = 1$ in our experiments to ensure a fair and consistent comparison.  We will clarify this point in the revised manuscript to avoid any potential misunderstanding.
>
> **W3:** We thank the reviewer for this valuable observation. The $O(n^2)$ space requirement primarily stems from storing graph-related structures such as adjacency or affinity matrices, rather than from the FRAM algorithm itself. To some extent, this is an inherent limitation of the graph matching problem. Nevertheless, this quadratic memory footprint is sufficient to handle the vast majority of practical graph matching tasks. Tackling significantly larger-scale problems may require fundamentally new paradigms—an important direction for future work, while it beyond the scope of this manuscript. Many classical methods incur even higher memory costs, such as those with $O(n^4)$ space complexity (see lines 261–262), demonstrating the relative efficiency of FRAM.
>
> **Q1**: We thank the reviewer for pointing out this oversight. We agree with the observation and will revise the complexity of Step 6 in Algorithm 1 to $O(Ln^2)$, where $L$ denotes the number of iterations as specified in Theorem 6. This revision ensures a more accurate characterization of the computational cost.
>
> **Q2:** We thank the reviewer for the helpful comment. The term _matching error_ is defined in Equation (18). To maintain consistency in terminology, we will revise the sentence on line 252 from
>
> > “For attributed graph matching tasks, we evaluate the accuracy of algorithms”
> > to
> > “For attributed graph matching tasks, we evaluate the performance of algorithms using the matching error defined in Equation (18).”
>
> This revision will help avoid confusion and ensure consistent usage of terminology throughout the paper.
>
> **Q3:** We thank the reviewer for pointing this out. In Algorithm 1, the matrix $K$ indeed refers to the product $FF^T$. We appreciate the reviewer’s suggestion and will explicitly introduce and clarify the definition of $K$ in Algorithm 1 in the revised manuscript to avoid any confusion.

---

> > ### Comment · Reviewer_2pmt · 2025-08-06
> >
> > Thanks for the response. I would increase my score.

---

> > > ### Author Response · Authors · 2025-08-07
> > > **Response to reviewer 2pmt**
> > >
> > > Thank you very much for your kind update and for increasing your score. I truly appreciate your time and constructive feedback throughout the review process.

---

### Note · Authors · 2025-08-11

We sincerely thank all reviewers for their constructive suggestions and positive feedback on our work, as well as the AC for guidance throughout the process. Here, we respectfully present a brief summary for your kind reference.

Key strengths highlighted by reviewers:

    Novel formulation & algorithm – Reformulates the projection step as an assignment and proposes the SDNS algorithm, substantially improving the algorithm in both accuracy and efficiency (2pmt, tDfZ).

    Theoretical soundness – Provides solid analysis with convergence guarantees, error bounds, and insights on the regularization parameter (2pmt, tDfZ, jPAJ).

    Effective implementation – Introduces an innovative mixed-precision scheme that enhances performance and is technically well-grounded (tDfZ, 6upw).

    Strong empirical results & clarity – Demonstrates leading accuracy and speed on real datasets, supported by a clear and well-structured presentation (2pmt, tDfZ, jPAJ).

Main concerns and our responses:

    Lack of discussion on the core parameter $\theta$ (2pmt): We conducted additional experiments and provided a detailed analysis of its impact on performance. The reviewer was satisfied with the results and subsequently raised the score.

    Cost and risk of mixed-precision computing (tDfZ): We clarified the precision settings for the two main computation modules and showed that mixed precision introduces negligible accuracy loss in our implementation, addressing the concern.

    Lack of discussion of the objective function and classical methods (6upw): We explained that the nature of the objective function is implicit domain knowledge and committed to expanding this discussion in the revision. We further clarified that we did not adopt classical methods for projection because it remains unclear whether they can support the entire matching algorithm in low precision. As this reviewer did not participate in the discussion phase, we cannot confirm whether the concern was resolved.

    Error analysis (jPAJ): Due to the analytical complexity of our method, we shows analysis of two simpler and weaker algorithm classes as proxies, thereby indirectly demonstrating our method’s superiority. The reviewer appeared to accept this explanation.

We will incorporate the improvements discussed into the revised manuscript to further strengthen and enrich the paper. Once again, we thank the reviewers and the AC for their valuable feedback and support throughout this process.

---

### Decision · Program_Chairs · 2025-09-17

**Decision:**

Accept (poster)

**Comment:**

This paper provides a novel heuristic for Quadratic Assignment Problem(QAP) by posing it as a Frobenius regularized Linear Assignment problem. A new algorithm SDSN is provided which can solve the posed problem. While there was all around appreciation for the proof of convergence of SDSN, the major concern is lack of discussion on other continuous relaxations of QAP and how to position this result in that context. At this point the paper can at best be recommended as borderline due to these concerns.
In case the paper is accepted the authors should implement all the changes, and more so have a detailed discussion of contrasting the proposed objective with existing relaxations.